# Response of biological productivity to North Atlantic marine front migration during the Holocene

David J. Harning[1,2], Anne E. Jennings[2], Denizcan Köseoğlu[3], Simon T. Belt[3], Áslaug Geirsdóttir[1], Julio Sepúlveda[2]

[1]Faculty of Earth Sciences, University of Iceland, Reykjavík, Iceland
[2]Institute of Arctic and Alpine Research, University of Colorado and Department of Geological Sciences, University of Colorado, Boulder, USA
[3]Biogeochemistry Research Centre, Plymouth University, Plymouth, UK

*Correspondence to*: David J. Harning (david.harning@colorado.edu)

**Abstract.** Marine fronts delineate the boundary between distinct water masses and, through the advection of nutrients, are important facilitators of regional productivity and biodiversity. As the modern climate continues to change the migration of frontal zones is evident, but a lack of information about their status prior to instrumental records hinders future projections. Here, we combine data from lipid biomarkers (archaeal isoprenoid glycerol dibiphytanyl glycerol tetraethers and algal highly branched isoprenoids) with planktic and benthic foraminifera assemblage to detail the biological response of the marine Arctic and Polar Front migrations on the North Iceland Shelf (NIS) over the last 8 ka. This multi-proxy approach enables us to quantify the thermal structure relating to Arctic and Polar Front migration, and test how this influences the corresponding changes in local pelagic productivity. Our data show that following an interval of Atlantic Water influence, the Arctic Front and its associated high pelagic productivity migrated south-eastward to the NIS by ~6.1 ka BP. Following a subsequent trend of regional cooling, Polar Water from the East Greenland Current and the associated Polar Front spread onto the NIS by ~3.8 ka BP, greatly diminishing local algal productivity through the Little Ice Age. Within the last century, the Arctic and Polar Fronts have moved northward back to their current positions relative to the NIS and helped stimulate the productivity that partially supports Iceland's economy. Our Holocene records from the NIS provide analogues for how the current frontal configuration and the productivity that it supports may change as global temperatures continue to rise.

## 1 Introduction

Marine fronts are the boundary that separate different water masses and are a globally ubiquitous feature in the oceans (Belkin et al., 2009). By nature, marine fronts are characterized by strong horizontal gradients of typically correlated water properties, such as temperature, salinity and nutrients. Some fronts influence the position of geostrophic currents that act as conduits for heat, salt and nutrient transport (i.e. density front), whereas others contribute to localized hot spots of primary productivity and biodiversity (i.e. convergent front) (Belkin, 2004; Belkin et al., 2009). Moreover, the combination of downwelling with

enhanced productivity at marine fronts can lead to increase sedimentation rates in underlying sediments, while interactions between surface waters and the overlying troposphere can generate wind and precipitation anomalies (Minobe et al., 2008; Small et al., 2008; Belkin et al., 2009 and references therein). Hence, marine fronts exert a broad and significant impact on various physical, chemical, and biological aspects of the global climate system. As the state of the density-driven oceanic circulation pattern in our near future remains uncertain (e.g. Rahmstorf et al., 2015), gaining an improved understanding of the dynamic nature of frontal systems over time will allow us to improve forecasts in these sensitive regions.

Marine fronts in the North Atlantic mark the boundary between warm and saline Atlantic Water that advects equatorial heat northward (Buckley and Marshall, 2015) and Arctic and Polar Water that transports low salinity water and drift ice southward from the Arctic Ocean (Fig. 1, Swift and Aagaard, 1981). The Arctic Front (AF) separates Arctic and Atlantic Water, whereas the Polar Front (PF) separates Arctic and Polar Waters. Today, most of the North Iceland Shelf (NIS) is influenced by Arctic Water (Fig. 1, Stefánsson, 1962), which is formed in the Iceland Sea by winter cooling of Atlantic Water and mixing with Polar Water (Våge et al., 2013). The dense Arctic water then sinks to intermediate depths and flows into the North Atlantic as North Atlantic Deep Water (Swift and Aagaard, 1981; Malmberg and Jonsson, 1997). Through the mediation of enhanced local primary productivity (Zhai et al., 2012), the proximity of the Arctic and Polar Fronts provide a habitat central to the distribution and migration of Icelandic capelin schools (Ólafsdóttir and Rose, 2012), which in turn support higher food-web predators, such as Atlantic cod, whales and seabirds (Piatt and Methven, 1992; Vilhjálmsson, 2002; Davoren et al., 2003). Given its sensitive position relative to the AF and PF, the NIS is strategically located to study past variations in these frontal systems (c.f. Giraudeau et al., 2004), which can provide context for future changes in North Atlantic oceanic heat transport and for the sustainability of Icelandic fisheries (Árnason and Sigfússon, 2012).

An extensive collection of literature exists for the NIS that provides important datasets on temperature, productivity and sea ice changes during the Holocene (last 11.7 ka BP) (see Supplemental Table S1). Although these proxy records are influenced by additional environmental variables (i.e. seasonality, depth habitat of biota and/or nutrients) that can result in discrepancies between paleoceanographic interpretations, they consistently partition the Holocene into three divisions (e.g. Kristjánsdóttir et al., 2017). The major transitions occur around 6.4 and 3.5 ka BP, with the interval in between reflecting the highest levels of nutrient availability and local primary productivity, presumably related to the local stabilization of the Arctic Front (Giraudeau et al., 2004; Cabedo-Sanz et al., 2016; Kristjánsdóttir et al., 2017). Here, we leverage new high-resolution lipid biomarker and foraminiferal assemblage records to explore how Holocene changes in positions of the AF and PF are expressed in paleoceanographic and paleoproductivity proxies on the NIS. Our analysis builds on previous work by targeting two, high sedimentation rate sediment cores that are strategically located on the western and eastern NIS to capture regional hydrographic gradients. The coupling of biogeochemical and traditional proxies with statistical analyses allows for more secure and detailed interpretations of paleoproductivity, temperature, and the position of the AF and PF during the Holocene.

## 2 Modern oceanography

The interactions between several water masses on the NIS results in a highly variable and complex hydrographic setting. Along the Denmark Strait, the North Iceland Irminger Current (NIIC) branches from the Irminger Current (IC) and transports warm (>4 °C) and saline (>35 ‰) Atlantic Water along northwest Iceland (Stefánsson, 1962). These Atlantic Waters then meet cold (<0 °C) and low salinity (<27.7 ‰) Polar Water and drift ice that is carried south along the coast of Greenland by the East Greenland Current (EGC), where the boundary between the EGC and IC constitutes the local PF (Fig. 1). NIIC Atlantic Water can cool and mix with Polar Water in the Iceland Sea to form the submerged and westward flowing North Iceland Jet (NIJ, Våge et al., 2011; Jónsson and Valdimarsson, 2012; Semper et al., 2019) and eastward flowing Iceland-Faroe Slope Jet (IFSJ, Semper et al., 2020) (Fig. 1). The East Iceland Current (EIC) branches off the EGC and carries cool Arctic water (1 to 4 ºC) formed in the Iceland Sea eastward along the NIS (Stefánsson, 1962). Although the EIC is generally ice free, it can occasionally advect EGC Polar Water and drift ice (c.f. Ólafsson, 1999). The boundary between the EIC and Atlantic Water is the local expression of the AF (Fig. 1). On the NIS, the water column becomes density stratified as the NIIC Atlantic Water converges with the Arctic/Polar Waters of the EIC. At the site of MD99-2269, NIIC Atlantic Water commonly forms the bottom water, but farther east at JR51-GC35 Atlantic Water is likely to occur as an intermediate water mass between Arctic Intermediate Water and surface waters of the EIC (http://www.hafro.is/Sjora).

Although sea ice is dominantly transported from east Greenland to the NIS by the EGC and EIC, local production can also occur in extreme cold years (Ólafsson, 1999). The proximity of mean April and August instrumental sea ice edges (1870-1920 CE) to the NIS (Fig. 1) means that subtle changes in sea ice advection along the EGC/EIC can have profound changes in local climate (Ólafsson, 1999) as sea ice limits the exchange of heat, gases and moisture between the ocean and atmosphere, in addition to insulating the colder polar atmosphere from the relatively warmer ocean during winter (Harning et al., 2019). Strong Polar Water advection to the NIS stratifies the water column and limits mixing of nutrients carried in Atlantic Water (i.e. phosphate, nitrate, and silica) for renewal in the photic zone, thus limiting primary productivity (Stefánsson and Ólafsson, 1991). In contrast, an Arctic Water surface stimulates a spring phytoplankton bloom, whereas the dominance of Atlantic Water extends the bloom into the summer resulting in higher total seasonal biomass (Thordardóttir, 1984; Zhai et al., 2012).

## 3 Materials and Methods

### 3.1 Marine sediment cores

Two marine sediment cores separated by ~200 km on the NIS were selected for this study in order to capture the modern oceanographic gradients and ocean current positions. Core MD99-2269 (66.63°N, 20.85°W, 365 mbsl) is located on the western NIS and under the primary influence of the NIIC Atlantic Water, whereas core JR51-GC35 (67.33°N, 16.70°W, 420 mbsl) lies on the eastern NIS and is mainly influenced by the EIC Arctic Water (Fig. 1). A variety of paleoceanographic datasets currently exist for MD99-2269 (Supplemental Table S1). Most of these studies rely on a securely-dated Holocene age

**Deleted:** it

**Deleted:** Marine sediment

**Deleted:** c

**Deleted:** marine sediment

**Deleted:** , which

model derived from 27 AMS [14]C dates, tephrochronology and paleomagnetic secular variation (Stoner et al., 2007), of which 20 AMS [14]C dates cover the upper 15 m of sediment (i.e. last 8 ka BP). Several similar corresponding datasets are also available for JR51-GC35 (Supplemental Table S1). The age model derived for JR51-GC35 records is based on 10 [14]C dates, of which 8 span the upper 3.5 m of sediment (i.e. last 8 ka BP, Bendle and Rosell-Melé, 2007). For this study's inter-site comparisons, we focus on the biomarker records (i.e. alkenones and highly-branched isoprenoids) that have been analysed in both cores (Fig. 2, Bendle and Rosell-Melé, 2007; Cabedo-Sanz et al., 2016; Kristjánsdóttir et al., 2017). Given that sedimentation rates are nearly linear at both locations over the last 8 ka, we do not correct proxy records for sediment flux. All subdivisions of the Holocene (e.g. Early, Middle, and Late) follow the recent formalization described in Walker et al. (2019).

Alkenones are long-chain ketones produced by coccolithophore algae (class: *Prymnesiophyceae*) in the photic zone of the water column that vary the number of unsaturation in response to sea surface temperature (SST) (Brassell et al., 1986; Prahl and Wakeham, 1987). We re-calculated SSTs from the original JR51-GC35 record (Bendle and Rosell-Melé, 2007) using the global $U^{K'}_{37}$ calibration (Müller et al., 1998)

$$(\text{Eq. 1}) \; SST = (U^{K'}_{37} - 0.044)/0.033$$

that removes the influence of $C_{37:4}$ alkenones, which dominate in low temperature environments (Prahl and Wakeham, 1987) and may be more susceptible to diagenesis (Müller et al., 1998). On the other hand, the previously published alkenone record from JR51-GC35 (Bendle and Rosell-Melé, 2007) used the $U^{K}_{37}$ index, which includes $C_{37:4}$ alkenones, and an Atlantic core top calibration (Rosell-Melé et al. 1995). Considering that the bloom and primary period of phytoplankton production occurs during early spring along the NIS (Zhai et al., 2012), we interpret the alkenone SST from both of our study sites to hold a spring seasonal bias.

In terms of highly-branched isoprenoids (HBIs), the $IP_{25}$ biomarker is a mono-unsaturated $C_{25}$ HBI alkene biosynthesized by certain Arctic sea ice diatoms (Brown et al., 2014; Belt, 2018), whereas HBI III and HBI IV are tri-unsaturated $C_{25}$ HBI alkenes biosynthesized by some open-water diatoms (e.g. *Rhizosolenia* spp.) in the Arctic (Rowland et al., 2001; Belt et al., 2017). Consequently, the presence of $IP_{25}$ indicates the occurrence of seasonal sea ice along the sea ice margin, whereas the presence of HBI III reflects highly productive Marginal Ice Zones (MIZ) characterized by open water phytoplankton blooms (Belt et al., 2015, 2019). The proportion of HBI III to HBI IV, reflected in the novel $T_{25}$ index, has been used to measure the occurrence of the spring phytoplankton bloom in the Barents Sea (Belt et al., 2019), although a recent investigation demonstrated that these relationships may be more complex in other (sub)-Arctic regions (Kolling et al., 2020). In addition to spring blooms, Rowland et al. (2001) demonstrated a systematic increase in the amount of HBI III relative to HBI IV (i.e. higher $T_{25}$) with increasing growth rate of *Rhizosolenia setigera* cultured at higher temperatures.

**3.2 Archaeal isoprenoid glycerol dibiphytanyl glycerol tetraethers**

We present new subsurface temperature (subT) proxy records for each sediment core based on the distribution of isoprenoid glycerol dibiphytanyl glycerol tetraethers (GDGTs). GDGTs are cell membrane lipids biosynthesized by marine ammonia

oxidizing Thaumarchaeota that can alter the degree of cyclization (number of cyclopentane moieties) in response to ocean temperature (Könneke et al., 2005; Schouten et al., 2013). On the NIS, variations in cyclization seem to mostly respond to winter and/or annual subT (0-200 m, Harning et al., 2019), and potentially to a lesser extent, ammonia oxidation rates (e.g. Hurley et al., 2016). Total lipids extracts (TLEs) were obtained from freeze-dried sediment subsamples (~2-3 g, $n$=53 for MD99-2269, $n$=18 for JR51-GC35) by ultrasonication using dichloromethane:methanol (2:1, v/v) at the University of Plymouth. TLEs were then suspended in hexane:isopropanol (99:1, v/v), sonicated, vortexed, and filtered using a 0.45 μm PTFE syringe filter. Prior to instrumental analysis, samples were spiked with 10 ng of the $C_{46}$ GDGT internal standard (Huguet et al., 2006). Isoprenoid GDGTs were identified and quantified via high performance liquid chromatography – mass spectrometry (HPLC-MS) a Thermo Scientific Ultimate 3000 HPLC system interfaced to a Q Exactive Focus Orbitrap-Quadrupole MS at the University of Colorado Boulder after Harning et al. (2019). We adopt the $TEX_{86}^L$ index (Kim et al., 2010) to quantify the distribution of GDGTs

$$(\text{Eq. 2}) \ TEX_{86}^L = \log \left( \frac{[GDGT-2]}{[GDGT-1]+[GDGT-2]+[GDGT-3]} \right)$$

and then convert to winter/annual subT based on a local Icelandic surface sediment calibration (Harning et al., 2019)

$$(\text{Eq. 3}) \ subT = 27.898(TEX_{86}^L) + 22.723$$

### 3.3 Highly branched isoprenoids

We expand the HBI datasets of Cabedo-Sanz et al. (2016) ($IP_{25}$ and HBI III) by reporting corresponding records of HBI IV, a geometric isomer of HBI III (Fig. S1). HBI IV is often co-produced by the same open-water diatoms that produce HBI III (e.g. *Rhizosolenia* spp., Rowland et al., 2001) and the combination of the two has recently been shown to be a useful predictor of spring phytoplankton blooms in the Barents Sea (Belt et al., 2019). Analysis of purified hydrocarbon fractions containing HBIs was carried out using gas chromatography–mass spectrometry (GC–MS) in full scan (FS) and selected ion monitoring (SIM) modes (Cabedo-Sanz et al., 2016). HBI IV was identified based on its characteristic GC retention index ($RI_{HP5MS}$ = 2091) and mass spectrum (Belt et al., 2000; Belt, 2018). HBI IV quantification was achieved by comparison of mass spectral responses of its molecular ion ($m/z$ 346) in SIM mode with those of the internal standard (9-OHD, $m/z$ 350) and normalized according to an instrumental response factor. HBI III and HBI IV datasets are then expressed through the following equation

$$(\text{Eq. 4}) \ T_{25} = \left( \frac{[III]}{[III]+[IV]} \right)/0.62$$

where $T_{25} \geq 1$ provides a qualitative proxy measure for the occurrence of spring phytoplankton blooms in the Barents Sea (Belt et al., 2019). However, as noted by Kolling et al. (2020), this may not necessarily be the case for other (sub)-Arctic regions, so instead we prefer to interpret the relative changes in $T_{25}$ in our NIS records as growth rate changes after earlier culture studies (Rowland et al., 2001).

### 3.4 Planktic and benthic foraminifera

The modern distributions of planktic and benthic foraminifera from marine surface sediment around Iceland provide a key baseline for interpreting local environmental changes (Johannessen et al., 1994; Rytter et al., 2002; Jennings et al., 2004). Based on this framework, previous downcore records from MD99-2269 have focused on certain indicator species at low- (benthic and planktic species, Justwan et al., 2008) and high-resolution (planktic species, Cabedo-Sanz et al., 2016). We expand these records by presenting high-resolution (multi-decadal) and complete assemblage counts from 273 planktic and 295 benthic foraminifera samples from MD99-2269. Although there are more benthic samples, all planktic samples are paired with benthic foraminifera assemblages making them directly comparable. The original high-resolution planktic records for *T. quinqueloba* and *N. pachyderma* from MD99-2269 (Cabedo-Sanz et al., 2016) have been recounted, which removes the high amplitude variability in the original dataset. The foraminifera counts in this study now represent the finalized dataset.

All sub-samples were prepared by wet sieving at 63 μm, 106 μm and 150 μm. Our dataset combines samples that had been air-dried at 35 °C after sieving and stored dry together with ones that were not dried (Cabedo-Sanz et al., 2016). The former sample set was wetted prior to wet splitting, and all samples were reanalysed for assemblages in a buffered solution. Planktic (>150 μm) and benthic (>106 μm) foraminiferal assemblages are expressed as percentages of the total planktic and total benthic population. In addition to individual foraminiferal species profiles, we also use our assemblage datasets to estimate bottom water temperature (BWT, benthic foraminiferal dataset) and sea surface temperature (SST, planktic foraminiferal dataset). Temperature estimates were quantified using North Atlantic transfer functions (BWT, Sejrup et al., 2004; SST, Pflaumann et al., 2003) and Weighted Averaging Partial Least Squares Regression (WAPLS) and Maximum Likelihood (ML) techniques for BWT and SST, respectively. Although the transfer functions assume a relationship with spring/summer temperature likely due to the seasonal flux of phytodetritus that heterotrophic foraminifera feed on, foraminifera productivity likely occurs at other times of the year, which results in annually integrated temperature records.

**3.5 Statistical analyses**

*Step 1:* To visualize long-term trends in proxy time series, we performed locally weighted smoothing (LOESS) to help minimize the influence from outliers and short-term variability while retaining persistent shifts. The smoothing criterion for each time series was selected automatically after optimization using generalized cross validation. Values along the resulting LOESS curves were extracted using a 60 yr timestep, representing a balance between both the arithmetic mean (~90 yr) and median (~30 yr) of first derivatives for each time series.

*Step 2:* To detect persistent step shifts within individual time series, we performed Sequential T-test Analysis of Regime Shifts (STARS) on the LOESS-smoothed data (Rodionov, 2004, 2006). The algorithm was tuned to detect regime shifts on millennial timescales by setting the cut-off length to 60, and we report the timing of shifts identified at the 95% confidence level ($p \leq 0.05$). The timing of regime shifts is best interpreted as approximate, as their exact timing and magnitude are affected by the chosen confidence level and cut-off length on centennial timescales (e.g. Rodionov, 2006; Seddon et al., 2014). Moreover, direct comparison of analyses for different proxies are also limited by the varying resolution of different proxies within a single core. As an example, for MD99-2269 $TEX_{86}^{L}$ and HBI records were analysed at 100-200 year resolution

whereas foraminifera assemblages and derived temperatures were analysed at 20-100 year resolution, which would allow the latter to have greater precision in determining regime shift timing Comparison of regime shift timing between sediment cores should be approached with caution due to their variable resolution, where JR51-GC35 features lower sampling density, and uncertainties in respective age models. Given the above limitations, we consider regime shift changes within the same core and between study sites as meaningful if they are within several centuries of each other.

> *Step 3:* To evaluate the relative similarity between MD99-2269 productivity and temperature proxy time series, we performed complete-linkage Agglomerative Hierarchical Clustering (AHC) using Dynamic Time Warping (DTW) distance as a curve shape-based dissimilarity. In order to preserve the influence of short-term variability while rectifying missing data, we avoided using LOESS smoothing and linearly interpolated the proxy records across all available horizons (600+ data points).

## 4 Results

### 4.1 Trends and regime shifts identified in existing proxy records

Alkenone-derived $U^{K'}_{37}$ spring SST records from MD99-2269 (Kristjánsdóttir et al., 2017) and JR51-GC35 (Bendle and Rosell-Melé, 2007) reveal different overall patterns of temperature variability during the last 8 ka. The record from MD99-2269 is relatively lower in absolute spring SSTs as well as in the magnitude of shorter-term variability. It features four regime shifts rather than five as found in JR51-GC35 (Fig. 2a-b). Both $U^{K'}_{37}$ records indicate that highest spring SSTs were achieved during the earliest portion of the records with the lowest spring SSTs occurring variably between ~4 and 2 ka BP (Fig. 2a-b). $IP_{25}$-derived sea ice records from both sediment cores (Cabedo-Sanz et al., 2016) show a similar pattern of low sea ice occurrence in the early Middle Holocene followed by generally increasing amounts of seasonal sea ice towards present (Fig. 2e-f). Three positive regime shifts are identified in $IP_{25}$ record from MD99-2269, whereas four positive regime shifts are identified in JR51-GC35 (Fig. 2e-f). HBI III records from both sediment cores (Cabedo-Sanz et al., 2016) show a similar pattern over the last 8 ka, with a period of relatively high concentrations between 6.1 and 3.8 ka BP compared to the periods before and after (Fig. 2g-h). Three regime shifts are identified in MD99-2269 whereas four are identified in JR51-GC35 (Fig. 2g-h).

### 4.2 Archaeal GDGTs

GDGTs are present above the detection limit in all sediment samples (Fig. S2). GDGT-0/crenarchaeol ratios around and below 2 (Fig. S2) indicate that the GDGT pool is not altered by methanogenic contributors (Blaga et al., 2009), and that Thaumarchaeota are the most likely producers of these GDGTs. The low standard error of the local Icelandic temperature calibration (±0.4 °C, Harning et al., 2019) increases our confidence in the observed changes in $TEX_{86}^L$-derived subT at both locations. For core MD99-2269, subT range from 6.7 to 3.1 °C and are subdivided by five regime shifts. The first regime shift indicates an increase in subT whereas the subsequent four indicate continual decreases in subT at this site (Fig. 2c). SubT at JR51-GC35 range from 4.6 to 2.2 °C, and in contrast to MD99-2269, only show four regime shifts, which all indicate progressive decreases in subT (Fig. 2d). Although the timing of highest subT varies between the two core locations (~6 and 5

---

**Deleted:** quality of chronological control

**Deleted:** select

**Deleted:** marine

**Deleted:** Similar to the $IP_{25}$ records,

**Deleted:** marine

**Deleted:** dominant

**Deleted:** -d

ka at MD99-2269 and by 8 ka in JR51-GC35), the timing of lowest subT occurs during the last 1 ka in both records (Fig. 2c-d).

### 4.3 $T_{25}$

HBI IV was present above the detection limit in all sediment samples (Fig. S3). The $T_{25}$ records for both cores show similar trends, with highest values achieved during the early Middle Holocene and progressively lower values towards present (Fig. 2i-j). For MD99-2269, the $T_{25}$ record exhibits four negative regime shifts, whereas for JR51-GC35, the $T_{25}$ record exhibits five

negative regimes shifts (Fig. 2i-j). The overall decreasing LOESS trends as well as the consistently negative regimes shifts in both records indicate a generally continuous decline in pelagic productivity over the last 8 ka (e.g. Belt at al., 2019). Further, given that HBI III is produced at a relatively higher proportion under conditions of more rapid growth in some diatoms (e.g. *R. setigera,* Rowland et al., 2001), these records also suggest a long-term decrease in diatom growth rates.

**4.4 Planktic and benthic foraminifera**

Seven planktic foraminiferal species were identified in the >150 μm size fraction and used for SST estimation (Fig. S4). Our reconstruction of planktic foraminifera assemblage and SST estimates include low- and high-resolution data before and after 8 ka BP, respectively (Fig. S4-S6). We focus on two planktic indicator species for our paleoceanographic discussion; *T. quinqueloba* and *N. pachyderma. T. quinqueloba* is an AF species (Johannessen et al., 1994; Volkmann, 2000; Pados and

310 Spielhagen, 2014) that shows highest abundance during the early Middle Holocene and lowest abundance during the Late Holocene, with four negative regime shifts (Fig. 3c). *N. pachyderma* is a Polar Water species (Johannessen et al., 1994; Jennings et al., 2004) that shows lowest abundances during the early Middle Holocene, and three positive shifts during the Late Holocene when its maximum abundance is achieved (Fig. 3d). In terms of foraminifera-reconstructed SST (S.E.=1.3 ºC), there is an overall trend of cooling throughout the last 8 ka from ~10 °C to ~3 °C, although SST from 8 to ~4 ka BP and from

315 ~3 ka BP to present are relatively more stable than between 4 and 3 ka BP (Fig. 4b). Four negative regime shifts are identified throughout the SST record, the most pronounced of which occurs at ~4 ka BP (Fig. 4b).

In the benthic dataset, we focus on the two most abundant species, *Cassidulina neoteretis* and *Cassidulina reniforme*, for paleoceanographic interpretations. Around Iceland, *C. neoteretis* is abundant on the western NIS where it is associated with NIIC Atlantic Water and dramatically decreases eastward where Atlantic Intermediate Water forms the bottom water (Rytter

et al., 2002; Jennings et al., 2004). *C. reniforme* is an Arctic species prevalent on the NIS but has its highest abundance in the low-salinity fjords of NW Iceland (Rytter et al., 2002; Jennings et al., 2004). In our records, *C. neoteretis* shows maximum abundances during the Middle and Late Holocene boundary (4.2 ka BP) and exhibits three regime shifts throughout the last 8 ka (Fig. 3f). *C. reniforme* peaks during the early Middle Holocene and has five regime shifts throughout the last 8 ka (Fig. 3g). We note that another dominant species is *Nonionella iridea,* which formed up to 20 % of the benthic fauna (Fig. S5), and we

included it in our statistical analyses to better understand its environmental preferences (see Section 5.2). Thirty-three of the 65 species in the Sejrup et al. (2004) transfer function are present in MD99-2269, although neither *N. iridea* nor the agglutinated

species were included (Sejrup et al., 2004). The BWT estimates are less variable than the foraminifera-based SST estimates, with temperatures generally falling between ~2.5 and 5 °C and maximum temperatures occurring at ~4 ka BP (S.E.=1.0 °C). Two positive regime shifts are identified in the Middle Holocene followed by two negative regime shifts during the Late Holocene, consistent with the general long-term pattern of warming from 8 to ~4 ka BP and cooling from ~4 ka BP to present (Fig. 4d).

### 4.5 Agglomerative Hierarchical Clustering analysis

The AHC dendrogram of DTW distances shows that the 11 selected proxy records from MD99-2269 successfully separate into four clusters (Fig. 5). The first grouping of $T_{25}$, foraminifera SST, and $TEX_{86}^{L}$ subT clusters proxies that are influenced by annual near surface to surface temperatures. The spring $U^{K'}_{37}$ SST belongs to its own cluster likely due to seasonality differences that result in a relatively smaller range of temperature changes as well as warming since ~2 ka BP (Fig. 4a-c). The third grouping of HBI III, *T. quinqueloba* and, *N. iridea,* and to a slightly lesser extent, HBI IV, suggests that this cluster reflects a separate but dominant environmental variable on the NIS, likely pelagic productivity (see Section 5.2 for discussion on *N. iridea*). Finally, the fourth grouping of $IP_{25}$, *N. pachyderma* and foraminifera BWT likely clusters cold proxy indicators, such as the sea ice biomarker $IP_{25}$ and planktic foraminifera species associated with Polar Water and bottom waters. The fact that foraminifera BWT clusters in the fourth grouping may relate to the relatively depressed temperatures in the early portion of the record, despite a negative trend from ~4 ka BP to present (Fig. 4d) that would be consistent with trends observed in the aforementioned near surface temperature and productivity proxies (Figs. 3 and 4). In any case, we note that the similarity between BWT and Polar Water indicators is weaker than that between $IP_{25}$ and *N. pachyderma*.

### 5 Discussion
### 5.1 Frontal proxies on the NIS

Modern distributions of *T. quinqueloba* suggest that the species feeds in the high nutrient and productive waters within the warmer margins of AFs (Volkmann, 2000; Pados and Spielhagen, 2014). In prior work from MD99-2269, Cabedo-Sanz et al. (2016) used Principal Component Analysis to show that HBI III and *T. quinqueloba* were related, along with biogenic $CaCO_3$, as proxies for surface productivity. Our AHC analysis, which uses the updated *T. quinqueloba* record, in addition to HBI IV and *N. Iridea,* reinforces the relationship previously observed between HBI III and *T. quinqueloba* time series on the NIS (Fig. 5). Given the known environmental preferences of *T. quinqueloba,* we suggest that the NIS diatom producers of HBI III and IV as well as benthic foraminifera *N. iridea* thrive in similar hydrographic conditions. Although we lack information on diatom sources in our study, the distribution of some HBI III-producing diatoms (i.e. *Rhizosolenia hebetata* f. *semispina*) does trace the AF's modern position (Oksman et al., 2019), and are important members of Arctic Water assemblages in the North Atlantic (Koç Karpuz & Schrader, 1990; Oksman et al., 2019). Moreover, the close relationship of the productivity cluster (blue) to the temperature clusters (purple and green) (Fig. 5) highlight the fact that SST and productivity are innately connected on the NIS as warmer Atlantic Water with sufficient vertical mixing often carry higher nutrient loads that stimulate primary production

(e.g. Thordardóttir, 1984; Zhai et al., 2012). Collectively, this evidence indicates that HBIs III and IV and *T. quinqueloba* track the migration of the AF along the NIS, which is further controlled by the temperature of various ocean currents.

**5.2 Environmental controls on benthic foraminifera *Nonionella iridea***

*N. iridea*, first identified in the South Atlantic (Heron-Allen and Earland, 1932), is an often overlooked or missed constituent in benthic faunal assemblages (e.g. Sejrup et al., 2004) due its small size (Jennings et al., 2004) and because its abundance is underestimated using dry analyses. Experimental work indicates that *N. iridea* may feed on seafloor phytodetritus and/or the associated suboxic-hypoxic bacterial populations that can develop, but only dominates the assemblage in response to pulsed phytodetritus delivery (Gooday and Hughes, 2002; Duffield et al., 2015). The wet picking techniques employed in our study allowed us to quantify downcore variations in *N. iridea* and better understand its environmental preferences and role in Icelandic benthic communities. *N. iridea* shows highest taxa abundance during the Middle Holocene, which features one negative regime shift. Lowest abundance occurred during the Late Holocene, which features one negative regime shift followed by one positive (Fig. 6). Overall, the structure of the *N. iridea*, HBI III and *T. quinqueloba* records are similar (i.e. decreasing toward present) and the timing of regime shifts between them is consistent (Fig. 6). This notion of similarity is further supported by group clustering in our AHC analysis (Fig. 5). Although future studies including modern foraminifera distributions that include this species are still needed, this statistical evidence suggests that *N. iridea*'s food supply may have been influenced by the presence of frontal systems and/or warmer waters. In other words, the enhanced pelagic productivity of the AF likely resulted in increased export of phytodetritus to the seafloor where it could be consumed by *N. iridea*.

**5.3 Migration of the Arctic Front and Polar Front**

**5.3.1 Northern distal Arctic Front (8 to 6.3 ka BP)**

In this interval, planktic foraminifera assemblages and the derived annual SST estimates indicate that NIIC Atlantic Water was entering the MD99-2269 site, whereas Arctic Intermediate Water occupied the lower depths as indicated by the benthic fauna and low BWT estimates (Fig. 4b-c and 7a). Coccolithophore assemblage data indicative of NIIC Atlantic Water from 8-7 ka BP from the same core (Giraudeau et al., 2004) support the warm planktic foraminifera SST reconstruction. Both of our HBI III records (Figs. 2g-h and 3a), as well as % *T. quinqueloba* (Fig. 3c), document low surface productivity until ~6.1 ka BP, which we interpret to reflect a northward position of the AF relative to the NIS (Fig. 7a). Although overall diatom biomass was low, high $T_{25}$ values indicate increased diatom growth rates (Fig. 3b), likely driven in part by higher temperatures (Fig. 5). In addition, total benthic and planktic foraminifera counts were highest in this interval (Fig. S6), which may also indicate a long ice-free season for production of some Arctic species. Although this interval of low phytoplankton productivity was previously interpreted to reflect strong convection in the Iceland Sea and the production of Arctic Intermediate Water (Giraudeau et al., 2004), we note that these interpretations are inconsistent with our planktic foraminifera assemblage dataset. However, given the relatively high orbitally-induced seasonality at this time (Berger and Loutre, 1991), the periodic production of cool and fresh North Icelandic Winter Water (Stefánsson, 1962) may have increased along the northern coastline. The

Formatted: Font: Italic
Formatted: Font: Italic
Formatted: Font: Italic
Formatted: Font: Italic
Formatted: Font: Italic
Formatted: Font: Italic
Formatted: Font: Italic
Formatted: Font: Italic
Formatted: Font: Italic
Formatted: Font: Italic
Deleted: 8 to 6.3 ka BP (
Deleted: n
Deleted: )
Deleted: diatom

potential contribution of these local waters along with the distal position of the AF would have dampened surface productivity for many frontal marine photosynthesizers (Fig. 7a).

Our datasets from MD99-2269 provide a comprehensive view of corresponding ocean temperatures on the western

NIS during this early Middle Holocene interval. $U^{K'}_{37}$ and foraminifera SST estimates show respectively high yet decreasing spring and annual surface temperatures (Fig. 4a-b). In contrast, our $TEX_{86}^L$-derived subT and foraminifera BWT records show depressed yet increasing temperatures (Fig. 4c-d). Hence, MD99-2269's temperature profile depicts a strong annual vertical temperature gradient during the Early-Middle Holocene (~6 ºC, Fig. 4e), resulting from stratification of Arctic Intermediate Water at lower depths and NIIC Atlantic Water at the surface. At the JR51-GC35 core site, we also observe similarly high and

stable SST and relatively lower corresponding subT (Fig. 2b and 2d). However, a lack of BWT estimates in this core prevents us from analysing the vertical temperature gradient on the eastern NIS. Comparison of $TEX_{86}^L$ subT temperature records between the western and eastern NIS show comparable absolute estimates (Fig. 4f), which likely reflects the invasion of NIIC Atlantic Water across the surface to subsurface of the entire NIS (Fig. 7a).

**5.3.2 Local stabilization of the Arctic Front (6.1 to 3.8 ka BP)**

At the beginning of this interval, planktic foraminifera indicate that NIIC Atlantic Water dominated the surface whereas benthic foraminifera show that Arctic Intermediate Water occupied the lower depths on the western NIS (Fig. 3). At ~6.1 ka BP, abrupt increases in HBI III and *T. quinqueloba* abundance from MD99-2269 indicate significantly enhanced pelagic primary production on the western NIS, which although variable, is sustained to ~3.8 ka BP (Fig. 3). Although the HBI III record from

430 JR51-GC35 shows similar trends, the overall increase in inferred pelagic productivity is smaller (Fig. 2h). We interpret this spatial variability to reflect that by 6.1 ka BP, the AF was positioned near MD99-2269 but never or rarely expanded eastward to JR51-GC35, and that NIIC Atlantic Water was less of a contributor to the JR51-GC35 site (Fig. 7b). The increase in certain dinoflagellate and coccolithophore algae species (e.g. *Coccolithus pelagicus*) have also been used to track the presence of the AF at site MD99-2269 from 6.2 to 3.5 ka BP (Giraudeau et al., 2004; Solignac et., 2006). Not only do the dinocyst and *C.*

435 *pelagicus* records further support the timing of AF presence around MD99-2269, they also reflect short-term oscillations between NIIC Atlantic Water and EIC Arctic Waters (Giraudeau et al., 2004; Solignac et al., 2006). Hence, although the AF was located around MD99-2269 in this interval, its precise location on the NIS was not static.

In terms of temperature, $U^{K'}_{37}$ and planktic foraminifera proxies reflect a decrease in spring and annual SST, respectively, beginning by ~5.3 ka BP followed by a second regime shift at ~4 ka BP (Fig. 4a-b). As nutrient availability was

440 presumably higher, diatom growth rate and cell division inferred from decreasing $T_{25}$ was likely hampered by these decreasing SSTs (Fig. 3b). In contrast to the SST records, $TEX_{86}^L$ subT on the western NIS shows anomalous warmth from ~6 to 5 ka BP (Fig. 4c). Interestingly, this peak subT warmth closely aligns with the initial increase of HBI III and *T. quinqueloba* and south-eastward migration of the AF over the western NIS at MD99-2269 (Fig. 2). As shown by the foraminifera fauna and temperature proxies, this frontal migration began to reduce the stratification of the water column (Fig. 4e), which may have

445 allowed warmer water to advect to lower depths where Thaumarchaeota live (Fig. 7b). Alternatively, if Thaumarchaeota

became stressed over increased competition for $NH_4^+$ with frontal primary producers, their ammonia oxidation rates could decrease, which in culture studies has been shown to increase GDGT-based temperatures (Hurley et al., 2016). If this is the case, the lack of increased $TEX_{86}^L$ subT changes at JR51-GC35 may potentially suggest a reduced competition for nutrients and reinforce reduced advection of NIIC Atlantic Water and a distal position of AF on the east NIS.

Following the regime shift at ~3.8 ka BP in the HBI III record from MD99-2269, the PF expanded southeast over the NIS (Fig. 7c). At the surface, this is reflected by decreases in spring and annual SST, subT, BWT (Fig. 4a-d), decreases in % *T. quinqueloba* at the expense of % *N. pachyderma*, lower diatom growth rates (decreasing $T_{25}$), increased $IP_{25}$-inferred sea ice (Fig. 3), as well as continued thermal destratification (Fig. 4e). However, beneath the surface, benthic fauna indicate the progressive submergence of the NIIC Atlantic Water under EIC Arctic Water (Fig. 3 and 7b). On the eastern NIS at JR51-GC35, subT and $IP_{25}$-derived sea ice records also continue to decrease and increase, respectively, although to a lesser extent than at MD99-2269 (Fig. 2). Combined with records of water mass distribution inferred from radiocarbon reservoir ages from benthic foraminifera (deep water, Eiríksson et al., 2004) and planktic foraminifera assemblage and their $\delta^{18}O$ (surface water, Eiríksson et al., 2000; Knudsen et al., 2004), these proxies collectively reflect progressive cooling and/or strengthening of EIC Arctic Water, rather than the shift in surface water source observed at MD99-2269 (i.e. NIIC to EIC) (Fig. 4f). Additional records of ocean temperature and iceberg rafting along the NIS indicate contemporaneous millennial-scale cooling during this interval (e.g. Moros et al., 2006; Jiang et al., 2015) likely linked to changes in oceanic circulation, such as AMOC slowdown (Thornalley et al., 2009) and/or changes in atmospheric circulation patterns (Orme et al., 2018).

One record that remains challenging to interpret during the Middle Holocene is the $U_{37}^{K'}$ SST at JR51-GC35. Bendle and Rosell-Melé (2007) connected the high amplitude variability of $U_{37}^{K'}$ SST (Fig. 2b) to the strength of North Atlantic Deep Water (NADW) formation and the Atlantic Meridional Overturning Circulation (AMOC) by comparison of $U_{37}^{K'}$ SST to NADW proxy records south of Iceland. Although the amplitude of JR51-GC35's $U_{37}^{K'}$ SST variability is inconsistent with our additional SST records, the timing of these SST changes is similar to those inferred from paired Mg/Ca-$\delta^{18}O$ measurements of planktic foraminifera south of Iceland that reflect the relative strength of the AMOC and Subpolar Gyre (Thornalley et al., 2009). However, if changes in the AMOC explain SST variability along the eastern NIS, it would be expected that similar variations would also be present in the MD99-2269 record on the western NIS. One possibility that may explain the contrasting observations is that the warm NIIC positioned over MD99-2269 was fluctuating to and from the JR51-GC35 core site in accordance with variability in AMOC strength (Fig. 7b). Given that local coccolithophore communities have changed during the Holocene and that the species during this interval favor these dynamic surface conditions (Giraudeau et al., 2004), it is also possible that the high-amplitude changes in JR51-GC35's $U_{37}^{K'}$ SST may relate to algal community changes associated with the varying EIC vs NIIC source water.

### 5.3.3 Southward migration of Arctic Front (3.8 ka BP to present)

Consistently low HBI III concentrations at both core sites during the last 3.8 ka suggest that the PF remained a local feature along the NIS during the Late Holocene (Fig. 2 and 7c). Further, planktic foraminifera (low % *T. quinqueloba* and high % *N.*

*pachyderma*) and rising concentrations of IP$_{25}$ in MD99-2269 (Fig. 3) suggest that the EIC was now advecting more Polar
Water relative to Arctic Water. Accordingly, there are regime shifts toward lower spring and annual SST and subT at ~2 to 1.3
ka BP (Fig. 4), and toward *N. pachyderma* dominance at ~1.6 ka BP (Fig. 3). Although the annual vertical temperature gradient
on the western NIS had decreased to <1 ºC from its early Middle Holocene maximum (Fig. 4e), Late Holocene cooling is
observed at all depths (Fig. 4). The low SSTs in combinations with low pelagic nutrient availability is likely responsible for
the lowest Holocene diatom biomass and growth rates observed in the HBI III and T$_{25}$ records, respectively (Fig. 2). These
observations fit well within the context of previous proxy research that has shown this period to be the coldest of the Holocene
on the NIS (Eiríksson et al., 2000; Andersen et al., 2004; Jiang et al., 2015). Moreover, the periodic advection of sea ice around
east Iceland and increase in % *T. quinqueloba* in southwest Iceland marine sediment cores show that the AF had migrated to
the southern coastline during the peak of this cooling from ~1760 to 1920 CE during the Little Ice Age (Jennings et al., 2001).

Several regional proxy records, such as U$^{K}_{37}$ in MD99-2269 and MD99-2266, have recently noted an upturn in the
derived spring/summer SSTs over the last two millennia (e.g. Moossen et al., 2015; Kristjánsdóttir et al., 2017). One possibility
is that this warming is related to regional changes in the North Atlantic Oscillation (Orme et al., 2018), a dominant atmospheric
circulation pattern of the North Atlantic (Hurrell et al., 2003), which has been suggested to have been in a prolonged negative
state over the last two millennia that would have favored a stronger East Greenland Current (Olsen et al., 2012). However, in
light of the continued decrease in northern hemisphere summer insolation (Berger and Loutre, 1991) and growth of Icelandic
ice caps that require local summer cooling (Larsen et al., 2011; Harning et al., 2016, 2018, 2020; Anderson et al., 2018, 2019;
Geirsdóttir et al., 2019), Cabedo-Sanz et al. (2016) suggested that this "warming" may have been driven by several other
factors. These include either 1) the presence of a thin low-salinity Polar Water lid that restricted mixing with deeper colder
water and the supply of nutrients and/or 2) a seasonal shift of alkenone production to later summer months following more
persistent spring sea ice. Our foraminifera-derived annual SST from MD99-2269 (Fig. 4b) and foraminifera assemblage data
(Fig. S4 and S5) suggest that despite possible changes in seasonality, cold and low salinity Polar Water formed the surface
waters even during the spring/summer. These results support both of the mechanisms proposed by Cabedo-Sanz et al.
(2016) and suggest that drawing conclusions about regional spring SST warming over the last 2 ka should be approached with
caution until the nuances of the individual proxies are better understood.

**5.4 Past and future controls on the Holocene migration of the Arctic and Polar Fronts**

The dominant climate forcing during the Holocene has been the first-order, orbitally-driven decrease in northern hemisphere
summer and annual insolation (Berger and Loutre, 1991). Our NIS SST records depict a similar first-order decrease (Fig. 4a-
b), consistent with the previously recognized insolation-driven reduction in North Atlantic northward heat transport around
Iceland (e.g. Andersen et al., 2004). The Holocene reduction in northward heat transport has also been modulated by centennial
to millennal- and centennial-scale changes in oceanic processes such as NADW formation (Oppo et al., 2003), the stability
and strength of the AMOC and Iceland-Scotland deep water overflow (Bianchi and McCave, 1999; Hall et al., 2004; Thornalley
et al., 2013), and dynamics of the Subpolar Gyre in the Labrador Sea (Thornalley et al., 2009; Moffa-Sánchez and Hall, 2017),

as well as changes in atmospheric circulation such as the North Atlantic Oscillation (e.g. Moossen et al., 2015; Orme et al., 2018). While these oceanic and atmospheric processes are certainly important for the short-term variability of North Atlantic oceanographic heat transport, we opt to focus on the long-term changes highlighted in our LOESS smoothed records and statistical analyses. In this regard, the progressive migration of the Arctic and Polar Fronts around Iceland suggest that NH insolation and the concomitant changes in temperature have remained the primary controls on overall frontal migration. Similar first-order movements of frontal systems in response to NH insolation have also been documented south of Iceland on the Reykjanes Ridge (Perner et al., 2018), on the East Greenland Shelf (Jennings et al., 2002) and in the Nordic and Barents Seas (Hald et al., 2007; Risebrobakken et al., 2010). Given that the past movement of the North Atlantic frontal systems is connected to regional temperature variability, our new records may provide useful analogues for ongoing anthropogenic warming around Iceland.

Following the end of the Little Ice Age in ~1920 CE and a century of first-order warming (e.g. Hanna et al., 2006), the AF and PF have returned to positions along Greenland and proximal to the NIS (Fig. 7d), similar to those during the warm early Middle Holocene (Fig. 7b). Accordingly, enhanced primary productivity (increased *T. quinqueloba*) is noted in recent NIS sediment records (Simon et al., 2020), that in addition to the position of the AF and PF straddling the NIS, may also be driven by increased advection of Atlantic Water (Jónsson and Valdimarsson, 2012) and/or freshwater discharge from the Greenland Ice Sheet (Perner et al., 2019). As global temperature continue to rise and melting of the Greenland Ice Sheet accelerates, freshening of North Atlantic surface waters is expected to continue. Although recent melt may help stimulate this productivity (Perner et al., 2019), there likely exists a threshold where too much freshening will restrict nutrient availability required for photosynthesis, as we observe during the Early Holocene (Fig. 7a). Further, the observed slowdown (Rahmstorf et al., 2015) and conceivable future shutdown of AMOC from Greenland melt (Bakker et al., 2016) may also result in a southward shift of North Atlantic frontal systems. Although more emprical and modeling research is needed to forecast future climate trajectories, our records highlight the sensitivity of the AF and PF to past temperature changes, which hold important implications for the local pelagic productivity and fishing industry that it supports today.

**6 Conclusions**

- We present new TEX$_{86}^{L}$-derived subT and HBI productivity records from the western and eastern NIS, as well as new high-resolution planktic and benthic foraminifera assemblage (and corresponding temperature) records from the western NIS; all covering the last 8 ka.
- The Arctic Front, a zone of intensified pelagic productivity (as indicated by HBI III and % *T. quinqueloba*), moderate phytoplankton growth rates and warmer waters, migrated south-eastward to the NIS by ~6.1 ka BP, with greater influence on the west than the east. By ~3.8 ka BP, the Arctic Front migrated south of the NIS, allowing for cold, Arctic/Polar Water associated with the Polar Front (as indicated by IP$_{25}$ and *N. pachyderma*) to dominate the NIS for the Late Holocene.

**Deleted:** possible

**Deleted:** ,

**Deleted:**

**Deleted:** has

**Deleted:** also

**Deleted:** its

**Deleted:** is

**Deleted:** manifested by

**Deleted:**

**Deleted:** and productivity variability in the past

**Deleted:** fossil fuel-driven

**Deleted:** http://vedur.is

**Deleted:** economies

**Deleted:** stabilized on

**Deleted:** (s)

- Vertical temperature gradients on the western NIS were largest during the early Middle Holocene and progressively decreased to the lowest temperature gradients during the Late Holocene. Longitudinal temperature gradients suggest that warmer NIIC Atlantic Water was more influential on the west compared to the eastern NIS where cooler EIC Arctic Water dominated.

- The Holocene migration of the AF and PF has been primarily controlled by first-order decreases in northern hemisphere insolation, but the productivity it supports is also sensitive to freshening of surface waters. The future balance between these two variables will shape how the local configuration of marine fronts in the North Atlantic will develop under continued regional warming.

**Data availability**

Data from this publication is available on the PANGAEA database.

**Author Contributions**

DJH conceived the study following discussions with AJ and STB. AJ led foraminifera analyses, DK extracted samples and performed statistical analyses, and DJH performed GDGT analyses. STB and JS provided access to analytical infrastructure, supervised biomarker analyses, and contributed to data interpretation. STB, ÁG and JS supported analytical expenses. DJH wrote the manuscript with contributions and discussion from all co-authors.

**Declaration of Competing Interests**

The authors declare that they have no conflict of interest.

**Acknowledgements**

We extend great thanks to the scientific parties and crews of the RRS *James Clark Ross* and R/V *Marion Dufresne II* cruises for initial sediment core collection, as well as the numerous authors of previous studies (and in particular Dr. John T. Andrews) who have contributed to both cores' comprehensive datasets and planted the seeds for continued research. We also thank Dr. Suzanne Maclachlan at the British Ocean Sediment Core Research Facility (BOSCORF), UK, for making sediments available from core JR51-GC35 and Dr. Patricia Cabedo-Sanz for providing us with HBI IV biomarker data. University of Colorado students Allyson Rugg, Ian Courtney, Kelly Curtis, Jessica Scherer and Kylie Smith counted the foraminiferal assemblages in MD99-2269, expanding on the initial assemblage data produced by Nancy Weiner. We are also grateful to Dr. Nadia Dildar from the Organic Geochemistry Lab at CU Boulder for analytical support.

**Financial support**

This study has been supported by the Icelandic Research Council (RANNIS) Grant of Excellence #141573-051 to ÁG and co-authors, as well as internal University of Colorado Boulder funds to JS.

Deleted: Upon publication data

Deleted: will be uploaded to

Deleted: During the review of the manuscript, data will be made available upon reasonable request to the authors.

Deleted: TLEs

**Review statement**

This paper was edited by Dr. Bjørg Risebrobakken and reviewed by Dr. Maciej Telesiński and one anonymous referee.

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

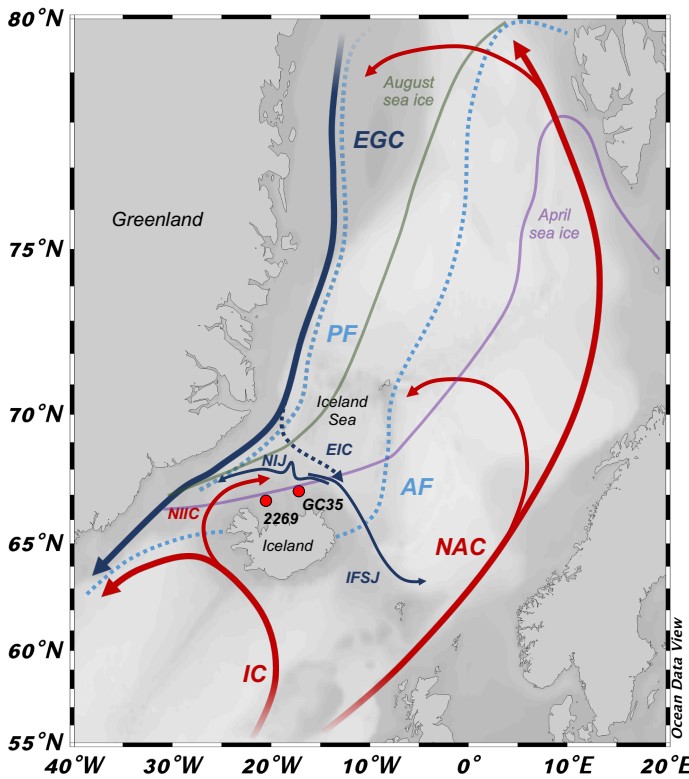

**Figure 1: Overview map.** Northern North Atlantic region with modern simplified position of ocean currents (bold lines), Arctic and Polar Fronts (dotted light blue lines, AF and PF, respectively), and mean 1870-1920 CE seasonal sea ice edges (April = purple, August = green, http://nsidc.org). Atlantic currents are red: IC = Irminger Current, NIIC = North Iceland Irminger Current, and NAC = North Atlantic Current. Polar currents are blue: EGC = East Greenland Current, NIJ = North Iceland Jet, and IFSJ = Iceland-Faroe Slope Jet. Arctic currents are dashed blue: EIC = East Iceland Current. Sediment core locations marked by red circles with abbreviated core names (i.e. MD99-2269=2269

and JR51-GC35=GC35). Base map from Ocean Data View (Schlitzer, 2020).

| Deleted: and |
| Deleted: Marine s |

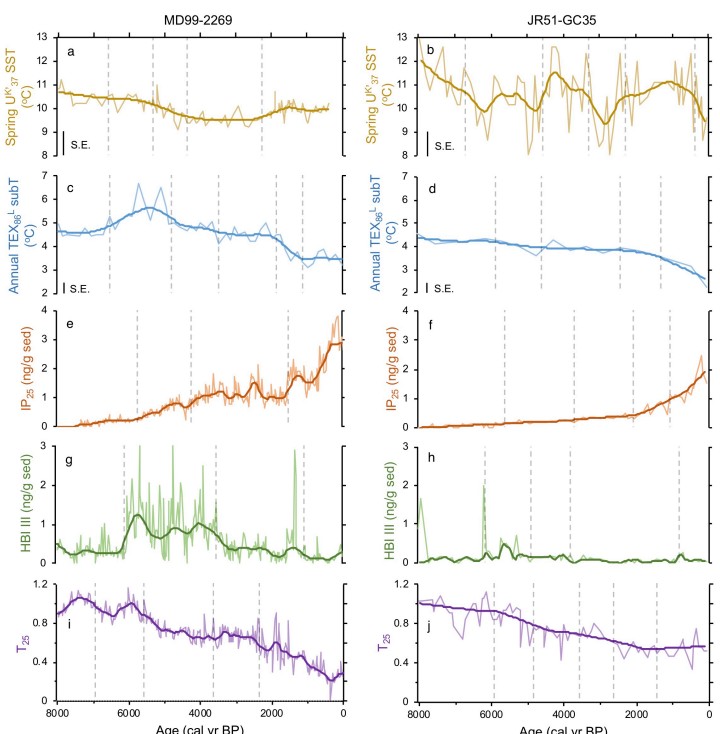

**965**    **Figure 2: Shared lipid biomarker records for MD99-2269 (left) and JR51-GC35 (right).** a-b) alkenone-derived SST (Bendle and Rosell-Melé, 2007; Kristjánsdóttir et al., 2017), c-d) GDGT-derived sub T (this study), e-f) IP$_{25}$ proxy for seasonal sea ice (Cabedo-Sanz et al., 2016), g-h) HBI III proxy for highly productive MIZ (Cabedo-Sanz et al., 2016), and i-j) T$_{25}$ proxy for spring phytoplankton bloom growth rates (this study). Y-axes have the same scale per proxy for easier visualization of proxy records between the two sites. Vertical dashed gray lines indicate regime shifts statistically identified in the LOESS-smoothed records.

**970**

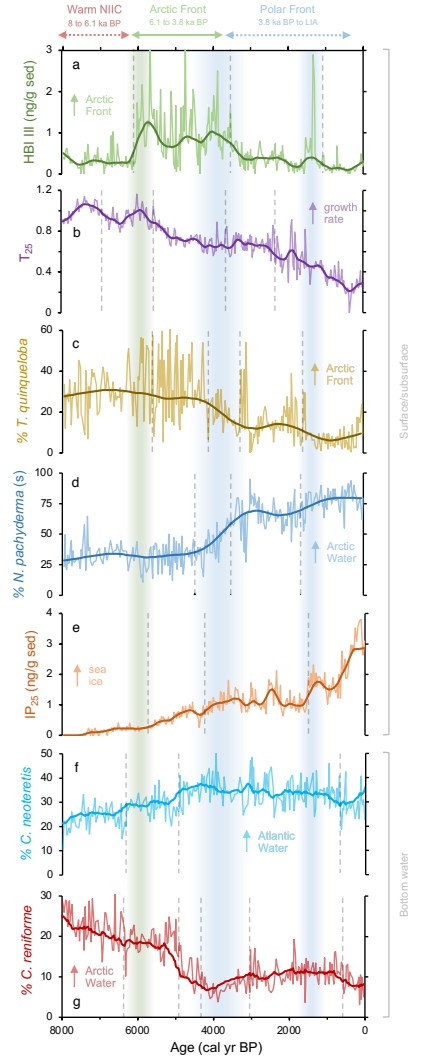

**Figure 3: Qualitative productivity and sea ice proxies from MD99-2269.** a) HBI III proxy for highly productive MIZ and marine fronts (Cabedo-Sanz et al., 2016), b) $T_{25}$ proxy for spring phytoplankton bloom growth rates (this study), c) % *T. quinqueloba* planktic foraminifera (this study), d) % *N. pachyderma* planktic foraminifera (this study), e) $IP_{25}$ proxy for seasonal sea ice presence (Cabedo-Sanz et al., 2016), f) % *C. neoteretis* benthic foraminifera (this study), and g) % *C. reniforme* benthic foraminifera (this study). Vertical dashed gray lines indicate regime shifts statistically identified in the LOESS-smoothed records. Vertical green bar indicates the timing of Polar Front establishment in the early Middle Holocene. The first blue bar (Middle Holocene) indicates the southward migration and departure of the Arctic Front from the NIS, and the second (Late Holocene) blue bar indicates a period of further cooling interpreted from the proxy regime shifts.

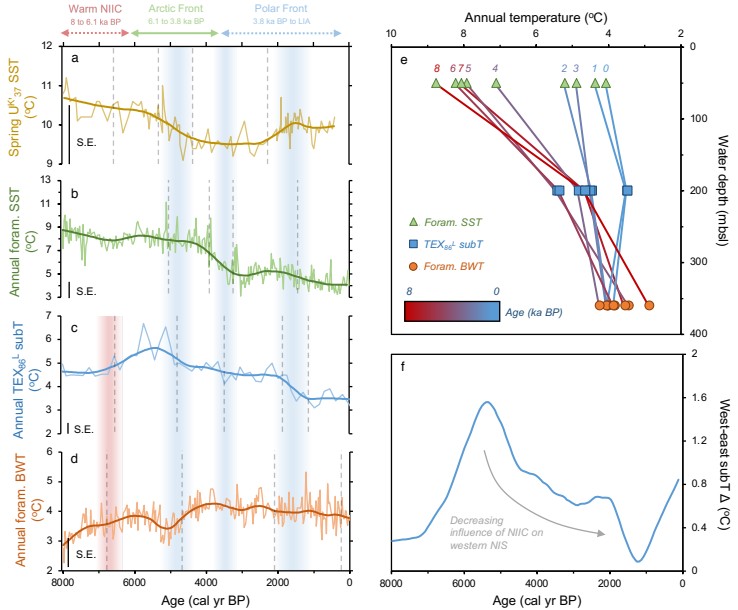

**Figure 4: Quantitative temperature proxies from MD99-2269.** a) $U^{K'}_{37}$ SST (Kristjánsdóttir et al., 2017), b) foraminifera assemblage SST (this study), c) $TEX_{86}^L$ subT (this study), and d) foraminifera assemblage BWT (this study). All panels show the corresponding calibration standard errors (S.E.) in the bottom left. Vertical dashed gray lines indicate regime shifts statistically identified in the LOESS-smoothed records. Vertical red bar indicates warming in subT (c) and BWT (d), whereas vertical blue bars indicate periods of progressive cooling expressed in proxy regime shifts. e) vertical temperature gradient inferred from MD99-2269 LOESS-smoothed annual temperature

proxies in 1000-year time slices (red-blue graded lines) and f) longitudinal subT gradient inferred from the difference between LOESS-smooth $TEX_{86}^L$ records on the west (MD99-2269) and east (JR51-GC35) NIS.

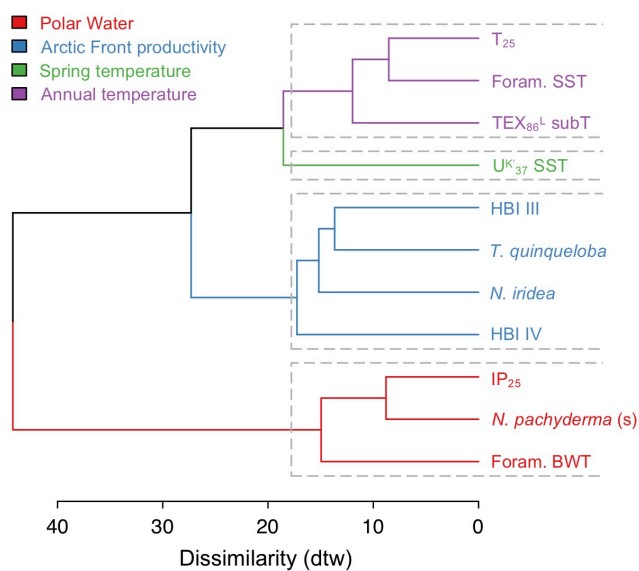

**Figure 5:** Four hierarchical clusters of select MD99-2269 productivity and temperature proxy records as determined by Dynamic Time Warping (DTW).

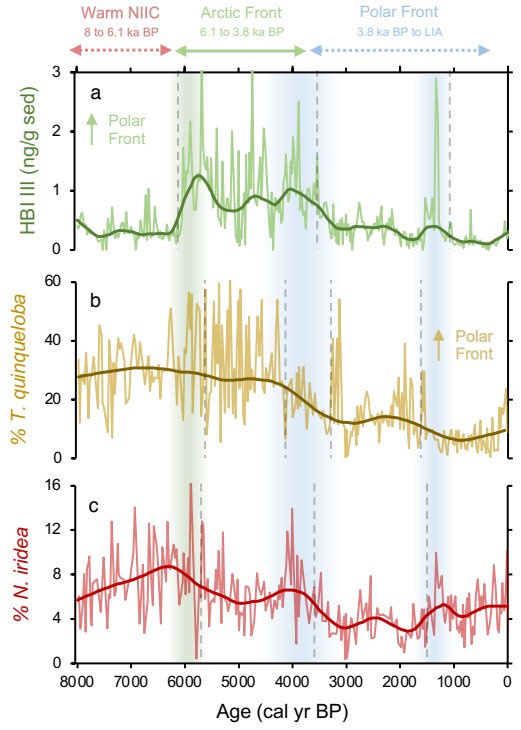

**Fig. 6: Comparison between frontal proxies from MD99-2269.** a) HBI III (Cabedo-Sanz et al., 2016), b) % *T. quinqueloba* planktic foraminifera (this study), and c) % *N. iridea* benthic foraminifera (this study). Vertical green bar indicates the timing of Polar Front establishment in the early Middle Holocene. The Middle Holocene blue bar indicates the southward migration and departure of the Polar Front from this location, and the Late Holocene blue bar indicates a period of further cooling interpreted from the proxy RSI values.

**Deleted:** marine sediment core

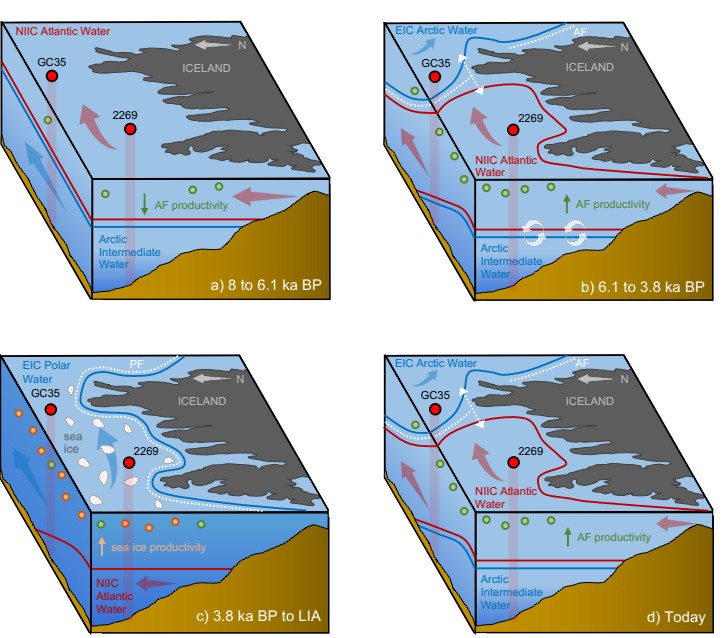

**Figure 7: Schematic illustration of NIS oceanography and migration of the Arctic (AF) and Polar Fronts (PF) at four Holocene time slices.** a) 8 to 6.1 ka BP, where the NIS is characterized by NIIC Atlantic Water at the surface with the AF and PF north of both core sites. b) 6.1 to 3.8 ka BP, where AF has migrated to the NIS and the NIIC Atlantic Water has become submerged beneath EIC Arctic Water on the east NIS. c) 3.8 ka BP to LIA, where the PF has migrated the NIS with EIC Arctic/Polar Water present throughout the surface and above NIIC Atlantic Water. d) Today, where the AF is present on the NIS with NIIC Atlantic Water to the west and EIC Arctic Water to the east. Also shown are simplified direction of ocean currents (blue and red arrows), open water and sea ice productivity indicators (green and orange circles, respectively), and sediment core locations and profiles (red circles and columns) with abbreviated core names (i.e. MD99-2269=2269 and JR51-GC35=GC35).

