# Peer review of "Response of biological productivity to North Atlantic marine front migration during the Holocene"

_Climate of the Past, 2020_

## Referee Comment (RC1) · Maciej M. Telesiński (Referee) · 21 Oct 2020

Harning et al. present an interesting and comprehensive study of the Mid-Late Holocene paleoceanography on the North Icelandic Shelf. They use a broad spectrum of micropaleontological and biogeochemical proxies seasoned with a sound statistical treatment. The major drawback of the manuscript is that some of the proxies were analyzed only in one core, which precludes a full comparison between the two records. It is also a pity that the records do not cover the entire Holocene. However, the study will certainly be of interest both for the researchers interested in the paleoceanographic/paleoclimatic history of Iceland and as a validation/comparison of the

proxies used. It also has some implications for the Icelandic economy. Therefore I find the MS suitable for publication in Climate of the Past after minor revisions according to general, specific and technical comments listed below. I am looking forward to the authors' response and further discussion.

Best regards, Maciej Telesiński

General comments

Abstract: The abstract is not very clear, especially for a reader unfamiliar with NIS. I suggest using terms like 'westward/eastward migration of the front' etc. 'the AF stabilized on the NIS' doesn't say much.

The regime shift analysis: Even though the regime shift analysis certainly has a sound statistical basis, it is not convincing for me. First of all, the number and timing of shifts are generally different in both records so they do not bring much to the discussion. You attempt to group them (vertical bars in Figs. 3 & 4) but it is not convincing either as the intervals are different for different sets of proxies (Fig. 3 vs. 4) and they do not agree with the boundaries between the intervals defined in the discussion. Even regime shifts within the same proxy (%N. pachyderma and %T. quinqueloba) do not agree.

The timing of major oceanographic shifts (and the discussed intervals) is unclear. The abstract and conclusions give 6.1 and 3.8 ka (lines 18-19 and 428-429), the introduction – 6.4 and 3.5 ka (line 54) and the discussion – 6.3 and 3.4 ka (lines 293 and 321). Please give consistent ages. Also, the naming of the 3 intervals is unclear. Are they identical with 'early MidH', 'late MidH' and 'LateH'?

If you write 'Middle/Late Holocene' (with capital letters) it implies that you use the formal subdivision of the Holocene. You should refer to Walker et al. 2019 JQS and use the ages given there as the boundaries. Otherwise use 'middle/late Holocene'.

Discussion: The discussion is filled with references to changes in individual proxies which is largely a repetition of the Results section and make the discussion hard to

follow. Please try to generalize the information from individual proxies and describe rather changes in environmental parameters and the implications for the oceanographic evolution.

Specific comments

37-40: Please rephrase. First, you write that the fronts separate AtlW and PW and then there appears another water mass between them. This might be confusing.

79: It is unclear what does 'it' refer to. Sea ice, I presume? Please rephrase.

88-89 and elsewhere: It is already stated in the previous sentence that these are 'marine sediment' cores so it is unnecessary to repeat it here and elsewhere.

106-112 and 132-144: The information on HBI III and IV is partly a repetition. Please rephrase.

152 and elsewhere: The '(s)' is unnecessary. As N. pachyderma and N. incompta are two different species it is obvious that by N. pachyderma you mean the left-coiling specimens.

161: Which technique was used for which reconstruction? Please specify.

184: Why are the results described in a different order than they are shown in Fig. 2? This is slightly confusing. Please reorder either the description or the figure.

194-196: This sentence is strangely formulated, suggesting that the IP25 and HBI III records show similar patterns. The similarity between the HBI III records in both cores is not evident, at least not in the figure with equal vertical scales. Please rephrase.

210: As the value of T25 = 1 seems to be an important threshold (line 142), the relation of the data to this value should also be described here.

236-237: I would rather say that C. neoteretis shows maximum abundances in the Late Holocene or the late Middle Holocene (after 5 ka).

254-256 and 258-263: Consider moving these sentences to the discussion.

265: The paragraph lacks the discussion on N. iridea and HBI IV which are grouped in the same AF cluster.

271-291: As you dedicate a considerable part of the discussion to N. iridea, consider adding the reference to its original description (Heron-Allen & Earland, 1932).

279: Please explain why the species' abundance is underestimated using dry analyses.

299: How can the percentage of T. quinqueloba document low surface diatom productivity? Please rephrase.

314: From Fig. 4e, I would say it is rather ∼6oC than ∼8oC.

322-323: Please rephrase to, e.g., "At the beginning of this interval, planktic foraminifera indicate that NIIC Atlantic Water dominated the surface whereas benthic foraminifera show that Arctic Intermediate Water occupied the lower depths on the western NIS (Fig. 3)."

324: At 6.1 ka the T. quinqueloba abundance decreases (at least the LOESS-smoothed line).

334-335: The regime shifts have different timing for the two proxies (see the general comment on regime shifts).

354-355: This is contradictory to what is shown in Fig. 4f (the figure shows a decreasing influence of NIIC, i.e., a shift in surface water source). Please rephrase or change/remove the figure.

360-361: Please explain how the changes in AMOC strength influence the SST.

386-397: For the discussion of the warming over the last two millennia I suggest taking into consideration the shift in NAO (Olsen et al. 2012 NatGeo).

399-421: The section does not discuss the controls on the Holocene migration of the

Arctic and Polar Fronts sufficiently. It rather gives an impression that all these state-of-the-art methods and the discussion are unnecessary if the frontal system is driven by insolation anyway. Furthermore, from Fig. 7 the fronts' migration does not seem to be progressive if the modern state is almost identical to that of the (late) Mid-Holocene (7b). Please expand the discussion in this section. Finally, the second paragraph could be a separate subsection (e.g., 'Future implications/predictions').

406: It is unclear what does 'its' refer to.

407-408: The ongoing warming is not driven by fossil fuels themselves, but by the burning of them. Please rephrase to, e.g., anthropogenic warming.

428: Can you really say that it stabilized if large variability both in HBI III and in %T. quinqueloba is observed over the 6.1-3.8 ka interval?

433-435: This conclusion is quite obvious just looking at the oceanographic setting of the NIS.

Fig. 7: What is the Early Holocene reconstruction based on if the presented records start in the early Middle Holocene? The diagrams should present the oceanographic conditions in intervals 8-6.3 ka, 6.3-3.4 ka and 3.4 ka – present (and perhaps 'present' based on modern oceanographic data for reference). See also the general comment on the timing of major oceanographic shifts.

Fig. S2: Please rephrase "GDGT-0/crenarchaeol values around and below 2 (grey dotted line) indicate minimal GDGT contributions from methanogenic archaea that may compromise TEX86L-based temperature inferences" to, e.g., "GDGT-0/crenarchaeol ratios around and below 2 indicate that the GDGT pool is not altered by methanogenic contributors"

Fig. S5: What is the blue vertical bar?

Technical corrections

11-12: Shouldn't it be "the migration of frontal zones"?

164: results

178: select – is this word correct here?

239: was included?

271-291: N. iridea in italics.

294 and subsequent subsection titles: Change to "Northern distal Arctic Front (8 to 6.3 ka BP)" and so on.

329: Change 'to' to 'at site'.

377: Remove 'from'.

409: Not sure if this is the correct way of citing a website in CP (please refer to the guidelines for authors) and if citing the Icelandic Met Office homepage is the most relevant here. Are there no papers describing the last century's warming?

Fig.1. The colours of the sea ice edge lines are hardly visible.

Fig. 2: Please indicate the meaning of individual proxies (as in Fig. 3 – Arctic Front, growth rate etc.)

Fig. 2i-j: The value of T25 = 1 seems to be an important threshold (line 142) so I suggest marking it on the plots with a dashed horizontal line.

Figs. 2, 3, 4a, 6: Please add ages on the upper axis for easier reading of the plots. Marking the intervals as defined in the discussion (8-6.3, 6.3-3.4, 3.4-present or 'early MidH', 'late MidH', 'LateH') would also be helpful.

Fig. 4e: Giving exact ages of individual lines would be more informative than the colour coding used (see a similar example in Fig. 10 of Hald et al. 2007 QSR)

Fig. 5: Consider changing colours – red somehow does not fit with Polar Water. E.g.,

blue – purple – green – red.

Fig. 6a: The regime shifts are not marked.

Fig. S4: Globigerinita glutinata, not qlutinata

Supporting Information Text S1 was not included in the Supplemental Material file and was therefore not reviewed.

---

## Referee Comment (RC2) · Anonymous Referee #2 · 3 Nov 2020

Harning and colleagues present new biomarker and foraminifera proxy data from two sediment cores from the North Iceland Shelf that are combined with previously published proxy data to investigate the oceanic front migrations and associated productivity changes during the last 8 ka. The multiproxy records show clear signals that are investigated also by means of statistical analyses. The various temperature, productivity and sea ice proxy results are well explained and allow an assessment of spatiotemporal changes in the positions of the Arctic Front and Polar Front during the Holocene. The manuscript is well written, clear and concise, and the figures are of high quality. Even though the major findings do not appear as surprising, the study will be of interest for a broad readership interested in North Atlantic/Nordic Seas paleoceanography and

paleoclimatology as well as in intercomparisons between different organic geochemical and micropaleontological proxy methods. Therefore, I find the manuscript suitable for publication in Climate of the Past.

I only have a few comments that might help improving the manuscript.

General comments:

The time intervals for which specific front positions and productivity regimes are reconstructed and discussed are inconsistent. For example, the intervals described in the abstract range from 8 to 6.1 ka and 6.1 to 3.8 ka, while the discussion is subdivided into intervals from 8 to 6.3 ka and 6.3 to 3.4 ka. This is confusing and should be revised. It would also be helpful to indicate the early, mid and late Holocene in the figures showing the time series. Also, the time intervals for the four Holocene time slices shown in Fig. 7 could be specified in each panel.

The statistically identified regime shifts are quite different in the various proxy records, probably related to the constraints of the statistical analysis as described in section 3.5. Would it be possible and useful to calculate regime shifts for multiple records combined?

The authors refer to first-order, orbitally driven decrease in northern hemisphere summer and annual insolation as the dominant forcing of the Holocene migration of the Arctic and Polar Fronts (lines 399-406). This section is rather short and it would be useful to explain in more detail how the decrease in insolation led to shifts in the ocean fronts. Furthermore, millennial-scale changes in NADW formation and AMOC strength are briefly mentioned, especially with respect to the SST variability in core JR51-GC35 during the Middle Holocene, but considered as not important control of the progressive migration of the fronts (lines 402-406). However, the frontal proxies in core MD99-2269, including %T. quinqueloba, HBI III and %N. iridea, do reflect millennial-scale variations with rather rapid changes, as depicted by several regime shifts in those records (Fig. 6). Could this not be seen as indication that the migration of the Arctic and Polar Fronts

also responded to millennial-scale changes in oceanic and/or atmospheric circulation, for example related to the subpolar gyre circulation? I feel that the discussion of the forcings in section 5.4 should be expanded a little bit.

Specific comments:

Line 18: Perhaps replace "stabilized" by "was located".

Line 20: Perhaps specify that the Arctic and Polar Fronts have moved back "northward" to their current positions.

Line 37: Revise "advect" ("advects").

Lines 95-96: The sedimentation rates and resulting temporal resolution of the proxy records could be mentioned here.

Lines 140-144: The link between the TR25 index and spring phytoplankton blooms as well as the equation for calculating the T25 were established based on samples from the Barents Sea only. A word of caution and a reference to Kolling et al. (2020) (Biomarker distributions in (sub)-Arctic surface sediments and their potential for sea ice reconstructions, G-cubed 21, https://doi.org/10.1029/2019GC008629) may be appropriate regarding the application of this proxy in other regions.

Line 178: "between select"; This sentence may be revised.

Line 278 and elsewhere: N. iridea should be in italics throughout the manuscript.

Line 352: Specify if "radiocarbon reservoir ages" were inferred for surface and/or deep water?

Line 377: "and from toward"; This sentence needs to be revised.

Fig. 1: Add a reference (Schlitzer, 2020) to the caption for the use of Ocean Data View.

In summary, the manuscript by Harning et al. is well written and contains high quality data that are potentially important new contributions to understanding natural ocean

climate variability. I would be happy to recommend publication of this manuscript in Climate of the Past, if the authors address the comments and questions.

---

## Author Comment (AC1) · 28 Nov 2020

*Harning et al. present an interesting and comprehensive study of the Mid-Late Holocene paleoceanography on the North Icelandic Shelf. They use a broad spectrum of micropaleontological and biogeochemical proxies seasoned with a sound statistical treatment. The major drawback of the manuscript is that some of the proxies were analyzed only in one core, which precludes a full comparison between the two records. It is also a pity that the records do not cover the entire Holocene. However, the study will certainly be of interest both for the researchers interested in the paleoceanographic/paleoclimatic history of Iceland and as a validation/comparison of the proxies used. It also has some implications for the Icelandic economy. Therefore I find the MS suitable for publication in Climate of the Past after minor revisions according to general, specific and technical comments listed below. I am looking forward to the authors' response and further discussion.*

*Best regards, Maciej Telesiński*

Dr. Telesiński,

Thank you very much for you thorough and encouraging review of our manuscript. We fully recognize the limitations of the datasets and, if time and funding permitted, would have certainly preferred to perform all analyses on both records and throughout the entire Holocene. Alas, this leaves room for continued work, which we hope to explore in future studies. Below we address each of your individual comments and provide tracked changes where relevant in the attached revised manuscript. Thanks again for you time and efforts with this peer review.

Kind regards, David Harning et al.

*General comments*

*Abstract: The abstract is not very clear, especially for a reader unfamiliar with NIS. I suggest using terms like 'westward/eastward migration of the front' etc. 'the AF stabilized on the NIS' doesn't say much.*
Thank you for highlighting that this was not very clear. We have now edited the abstract to describe the fronts as migrating, rather than stabilizing, and with directional descriptors. We hope this will now help future readers.

*The regime shift analysis: Even though the regime shift analysis certainly has a sound statistical basis, it is not convincing for me. First of all, the number and timing of shifts are generally different in both records so they do not bring much to the discussion. You attempt to group them (vertical bars in Figs. 3 & 4) but it is not convincing either as the intervals are different for different sets of proxies (Fig. 3 vs. 4) and they do not agree with the boundaries between the intervals defined in the discussion. Even regime shifts within the same proxy (%N. pachyderma and %T. quinqueloba) do not agree.*
The regime shift analyses certainly have their limitations as we point out in the Methods and Materials Section 3.5. Most importantly, they should be interpreted as approximate within several 100 years on Holocene time scales. (e.g. Seddon et al., 2014). Within the same core, you are correct that the regime shifts do not align perfectly, which in our opinion, is not surprising as there are certainly additional environmental variables that influence each proxy records beyond what we interpret is the dominant variables (e.g. frontal productivity, temperature, and sea ice). With this in mind, we would argue that there are striking similarities between the timing of regime shifts, especially if the prior boundaries of 6.4 and 3.5 ka BP are considered (Kristjánsdóttir et al., 2017). Moreover, given the limitations we've pointed out, we have tried to not put full emphasis on these for paleoceanographic interpretations, but instead, use them more as a guide. We hope this clarifies any reservations you have about these statistical analyses.

*The timing of major oceanographic shifts (and the discussed intervals) is unclear. The abstract and conclusions give 6.1 and 3.8 ka (lines 18-19 and 428-429), the introduction – 6.4 and 3.5 ka (line 54) and the discussion – 6.3 and 3.4 ka (lines 293 and 321). Please give consistent ages. Also, the naming of the 3 intervals is unclear. Are they identical with 'early MidH', 'late MidH' and 'LateH'?*
We apologize for the inconsistency in the boundaries ages. Those based on our study should have been 6.1 and 3.8 ka BP, which has now been corrected in the manuscript. The boundaries of 6.4 and 3.5 ka BP

mentioned in the Introduction are based on a previous study that used different statistical techniques and proxy records (Kristjánsdóttir et al., 2017). The point of mentioning it there is to simply highlight that prior research from the NIS has found similar timing of boundaries between distinct paleoceanographic conditions.

*If you write 'Middle/Late Holocene' (with capital letters) it implies that you use the formal subdivision of the Holocene. You should refer to Walker et al. 2019 JQS and use the ages given there as the boundaries. Otherwise use 'middle/late Holocene'.*

Thank you for pointing this out. We were following the formal subdivision of Walker et al. (2019) and have now acknowledged this in the discussion of the age models in Section 3.1., and added to the reference list.

*Discussion: The discussion is filled with references to changes in individual proxies which is largely a repetition of the Results section and make the discussion hard to follow. Please try to generalize the information from individual proxies and describe rather changes in environmental parameters and the implications for the oceanographic evolution.*

We have edited the Discussion some for smoothing but prefer to keep the results integrated in the discussion as it mostly was. While we do mention some of this in the results, the proxies are not discussed together until the discussion, which is where we can actually derive interpretations of water mass source, temperature and productivity changes. We do provide generalized statements about oceanographic conditions throughout the discussion, as well as refer to Fig. 7, which we intend to provide a simplified description of the oceanographic evolution that we focus on in this paper.

*Specific comments*

*37-40: Please rephrase. First, you write that the fronts separate AtlW and PW and then there appears another water mass between them. This might be confusing.*

This is a good suggestion and agree that this may come off as confusing. We have reworded these sentences to improve the flow and clarity on the modern frontal and current systems around Iceland.

*79: It is unclear what does 'it' refer to. Sea ice, I presume? Please rephrase.*

'It' edited to 'sea ice'.

*88-89 and elsewhere: It is already stated in the previous sentence that these are 'marine sediment' cores so it is unnecessary to repeat it here and elsewhere.*

Good suggestion. Subsequent mentioning of 'marine sediment' cores have now been removed.

*106-112 and 132-144: The information on HBI III and IV is partly a repetition. Please rephrase.*

This is a good point, and the information has now been concentrated in the former Section 3.1. where we discuss the proxies and what information they provide for paleoenvironmental interpretations, rather than in the methods.

*152 and elsewhere: The '(s)' is unnecessary. As N. pachyderma and N. incompta are two different species it is obvious that by N. pachyderma you mean the left-coiling specimens.*

We have now removed the '(s)' to eliminate the redundancy, thank you.

*161: Which technique was used for which reconstruction? Please specify.*

WAPLS for BWT and ML for SST. Now specified in text.

*184: Why are the results described in a different order than they are shown in Fig. 2? This is slightly confusing. Please reorder either the description or the figure.*

We prefer to keep the structure in the paper as it is so as to describe the previously published records first, and then introduce the new records. We also prefer to keep the order of the records in Fig. 2, with surface temperature above subT to reflect the general structure of the water column but separated from the productivity proxies below. In both cases (temperature and productivity) we added new proxy records

(e.g. TEX86 and T25), which if reordered to separate the old and new, may be confusing. We hope this is not a significant issue and that the structure can remain the same, thank you.

*194-196: This sentence is strangely formulated, suggesting that the IP25 and HBI III records show similar patterns. The similarity between the HBI III records in both cores is not evident, at least not in the figure with equal vertical scales. Please rephrase.*
We have rephrased the sentence so as to eliminate the reference to IP25 that may introduce confusion.

*210: As the value of T25 = 1 seems to be an important threshold (line 142), the relation of the data to this value should also be described here.*
This is an important point to bring up. We have since clarified in the text that T25 values over 1 only indicate the occurrence of spring phytoplankton blooms in the Barents Sea where the proxy was originally developed (Belt et al., 2019). A recent study by Kolling et al. (2020) has since found that the T25 proxy does not relate as simply to spring phytoplankton blooms in other sub-Arctic regions, such as Baffin Bay. Hence, in our paper, we prefer to use the T25 proxy as an indicator of growth rates changes following some earlier culture studies by Rowland et al. (2001).

*236-237: I would rather say that C. neoteretis shows maximum abundances in the Late Holocene or the late Middle Holocene (after 5 ka).*
We agree with this interpretation, and have now corrected the text to state that the maximum abundances occurred at the Middle and Late Holocene boundary (4.2 ka B).

*254-256 and 258-263: Consider moving these sentences to the discussion.*
We agree with this suggestion and have moved it to the discussion on Frontal Proxies. We think this consolidation will also help improve our arguments at the end of the manuscript where we discuss the past and future controls on frontal productivity around Iceland, thank you.

*265: The paragraph lacks the discussion on N. iridea and HBI IV which are grouped in the same AF cluster.*
This is a great point, and we have now included both in the discussion.

*271-291: As you dedicate a considerable part of the discussion to N. iridea, consider adding the reference to its original description (Heron-Allen & Earland, 1932).*
This is certainly appropriate so we have now added the citation to the discussion of *N. iridea*.

*279: Please explain why the species' abundance is underestimated using dry analyses.*
*Nonionella iridea* is a fragile, often small species. In the past it was common practice to freeze dry samples prior to sieving and then to count the dry sieved fractions. Our original set of samples from this core (at 50 cm spacing) used that method and we did not report *N. iridea* in the samples. We can suggest two problems in MD99-2269 that cause the loss of *N. iridea* in dry samples. First, in the intervals with N. iridea there is also quite a bit of biological material that forms mats as the sample dries. The small fragile forams become entrained in the biological mats. Wetting the samples allows all of the forams to be counted as they are released from the mats. Secondly, the drying and wetting and drying of the samples stresses the thin-walled forams causing them to break. It also causes underrepresentation of agglutinated forams.

*299: How can the percentage of T. quinqueloba document low surface diatom productivity? Please rephrase.*
The inclusion of 'diatom' was an error, as we intended to simply refer to general surface productivity.

*314: From Fig. 4e, I would say it is rather ~6oC than ~8oC.*
Yes, that was indeed a typo. Good catch, thanks!

*322-323: Please rephrase to, e.g., "At the beginning of this interval, planktic foraminifera indicate that NIIC Atlantic Water dominated the surface whereas benthic foraminifera show that Arctic Intermediate Water occupied the lower depths on the western NIS (Fig. 3)."*

Good suggestion, thank you.

*324: At 6.1 ka the T. quinqueloba abundance decreases (at least the LOESS-smoothed line).*
We intended to refer to the raw records, which we have now clarified in the main text.

*334-335: The regime shifts have different timing for the two proxies (see the general comment on regime shifts).*
Please see our reply to your general comment.

*354-355: This is contradictory to what is shown in Fig. 4f (the figure shows a decreasing influence of NIIC, i.e., a shift in surface water source). Please rephrase or change/remove the figure.*
We apologize for any confusions and have attempted to clarify our intended meaning in the text. Fig. 4f describes the source water on the western NIS (i.e. MD99-2269), whereas in the text you refer to we are discussing conditions on the eastern NIS (i.e. JR51-GC35).

*360-361: Please explain how the changes in AMOC strength influence the SST.*
Bendle and Rosell-Melé (2007) connected their UK37 SST record to NADW formation via comparison to marine proxy records of NADW south of Iceland. This has now been clarified in the text.

*386-397: For the discussion of the warming over the last two millennia I suggest taking into consideration the shift in NAO (Olsen et al. 2012 NatGeo).*
While we only eluded to the NAO by referring to Orme et al. (2018), we acknowledge that this should have been more explicit. We have now expanded this section slightly to explain the variability in the NAO and how that would impact the local oceanographic conditions.

*399-421: The section does not discuss the controls on the Holocene migration of the Arctic and Polar Fronts sufficiently. It rather gives an impression that all these state-of-the-art methods and the discussion are unnecessary if the frontal system is driven by insolation anyway. Furthermore, from Fig. 7 the fronts' migration does not seem to be progressive if the modern state is almost identical to that of the (late) Mid-Holocene (7b). Please expand the discussion in this section. Finally, the second paragraph could be a separate subsection (e.g., 'Future implications/predictions').*
We respectfully argue that although it does not come as a surprise, without our presented datasets the community would not be able to conclude that millennial-scale changes in the migration of North Atlantic marine frontal systems are predominantly driven by NH summer insolation and temperature. We have expanded the text slightly to more explicitly acknowledge that although there are certainly other controls on higher frequency changes (e.g. NADW, AMOC, NAO), the overarching goal of this paper was to explore the long-term changes reflected in our LOESS-smoothed records. Future work is needed to understand the higher frequency changes in these records and their relation to other atmospheric and oceanic circulation patterns, which we hope to pursue at a later stage. Finally, we have edited the name of the section to 'Past and future controls on the migration of the Arctic and Polar Fronts'.

*406: It is unclear what does 'its' refer to.*
'It' refers to front migration. Now edited in the text.

*407-408: The ongoing warming is not driven by fossil fuels themselves, but by the burning of them. Please rephrase to, e.g., anthropogenic warming.*
Edited.

*428: Can you really say that it stabilized if large variability both in HBI III and in %T. quinqueloba is observed over the 6.1-3.8 ka interval?*
Similar to the abstract, we have modified this to state that the fronts 'migrated south-eastward' to the NIS rather than 'stabilized'. Thank you for highlighting this.

*433-435: This conclusion is quite obvious just looking at the oceanographic setting of the NIS.*
Thank you.

*Fig. 7: What is the Early Holocene reconstruction based on if the presented records start in the early Middle Holocene? The diagrams should present the oceanographic conditions in intervals 8-6.3 ka, 6.3-3.4 ka and 3.4 ka – present (and perhaps 'present' based on modern oceanographic data for reference). See also the general comment on the timing of major oceanographic shifts.*

This is a great point also brought up by Reviewer 2. In short you are correct that the first panel was not truly Early Holocene as we are following the formalized subdivisions by Walker et al. (2019). We have not edited the 4 panels to simply reflect the time boundaries that are discussed in the paper, that we clarified in your general comment on the timing of major paleoceanographic changes.

*Fig. S2: Please rephrase "GDGT-0/crenarchaeol values around and below 2 (grey dotted line) indicate minimal GDGT contributions from methanogenic archaea that may compromise TEX86L-based temperature inferences" to, e.g., "GDGT-0/crenarchaeol ratios around and below 2 indicate that the GDGT pool is not altered by methanogenic contributors"*

Rephrased as suggested.

*Fig. S5: What is the blue vertical bar?*

The blue bar was not intended to be there, so it has now been removed.

*Technical corrections*

*11-12: Shouldn't it be "the migration of frontal zones"?*

Correct, and edited accordingly.

*164: results*

Edited.

*178: select – is this word correct here?*

'Select' has been removed.

*239: was included?*

Edited to clarify our meaning, thanks.

*271-291: N. iridea in italics.*

Edited, thank you.

*294 and subsequent subsection titles: Change to "Northern distal Arctic Front (8 to 6.3 ka BP)" and so on.*

Edited, thank you.

*329: Change 'to' to 'at site'.*

Edited.

*377: Remove 'from'.*

'From' changed to 'at'.

*409: Not sure if this is the correct way of citing a website in CP (please refer to the guidelines for authors) and if citing the Icelandic Met Office homepage is the most relevant here. Are there no papers describing the last century's warming?*

We have edited the reference to the IMO to Hanna et al., 2006. Please see edited text and reference list in the main manuscript.

*Fig.1. The colours of the sea ice edge lines are hardly visible.*

We have made the sea ice edge lines less transparent in order to improve their visibility.

*Fig. 2: Please indicate the meaning of individual proxies (as in Fig. 3 – Arctic Front, growth rate etc.)*

Edited.

*Fig. 2i-j: The value of T25 = 1 seems to be an important threshold (line 142) so I suggest marking it on the plots with a dashed horizontal line.*

As we discussed in your specific comment, this threshold is really only of value in the Barents Sea where the proxy has been calibrated. Hence, we prefer not to add it in so as to reduce potential confusion for readers.

*Figs. 2, 3, 4a, 6: Please add ages on the upper axis for easier reading of the plots. Marking the intervals as defined in the discussion (8-6.3, 6.3-3.4, 3.4-present or 'early MidH', 'late MidH', 'LateH') would also be helpful.*

As originally we only had descriptors for along the x-axes (e.g. warm NIIC, Arctic Front, and Polar Front), we have now also added the time ranges for each as well. This is a great suggestion, thank you.

*Fig. 4e: Giving exact ages of individual lines would be more informative than the colour coding used (see a similar example in Fig. 10 of Hald et al. 2007 QSR).*

This is a good point, and we have now added the age (ka) in addition to the color grading, which we hope improves clarity for the read.

*Fig. 5: Consider changing colours – red somehow does not fit with Polar Water. E.g., blue – purple – green – red.*

Edited.

*Fig. 6a: The regime shifts are not marked.*

Regime shifts now added, thank you.

*Fig. S4: Globigerinita glutinata, not qlutinata*

Edited, thank you.

*Supporting Information Text S1 was not included in the Supplemental Material file and was therefore not reviewed.*

This was an error in the text, where Supporting Information Text S1 should have referred to Section 5.2. This has now been corrected.

---

## Author Comment (AC2) · 28 Nov 2020

*Harning and colleagues present new biomarker and foraminifera proxy data from two sediment cores from the North Iceland Shelf that are combined with previously published proxy data to investigate the oceanic front migrations and associated productivity changes during the last 8 ka. The multiproxy records show clear signals that are investigated also by means of statistical analyses. The various temperature, productivity and sea ice proxy results are well explained and allow an assessment of spatiotemporal changes in the positions of the Arctic Front and Polar Front during the Holocene. The manuscript is well written, clear and concise, and the figures are of high quality. Even though the major findings do not appear as surprising, the study will be of interest for a broad readership interested in North Atlantic/Nordic Seas paleoceanography and paleoclimatology as well as in intercomparisons between different organic geochemical and micropaleontological proxy methods. Therefore, I find the manuscript suitable for publication in Climate of the Past.*

Reviewer 2,

Thank you very much for your time and effort reviewing our manuscript. Below we address each of your individual comments and provide tracked changes where relevant in the attached revised manuscript.

Kind regards, David Harning et al.

*I only have a few comments that might help improving the manuscript.*

*General comments:*

*The time intervals for which specific front positions and productivity regimes are reconstructed and discussed are inconsistent. For example, the intervals described in the abstract range from 8 to 6.1 ka and 6.1 to 3.8 ka, while the discussion is subdivided into intervals from 8 to 6.3 ka and 6.3 to 3.4 ka. This is confusing and should be revised. It would also be helpful to indicate the early, mid and late Holocene in the figures showing the time series. Also, the time intervals for the four Holocene time slices shown in Fig. 7 could be specified in each panel.*

Thank you for highlighting this inconsistency! The boundaries should be 6.1 and 3.8 ka based our data analysis presented in this manuscript and have now been edited accordingly. The 6.4 and 3.5 ka boundaries mentioned in the introduction are after Kristjánsdóttir et al. (2017), and we only point them out to highlight that previous research has found similarly time boundaries that separate distinct paleoceanographic conditions. We have also added these time intervals to the four time slices in Fig. 7 to improve clarity. As Reviewer 1 noted, the earliest time interval is not actually "Early Holocene" so referring to specific dates is preferrable.

*The statistically identified regime shifts are quite different in the various proxy records, probably related to the constraints of the statistical analysis as described in section 3.5. Would it be possible and useful to calculate regime shifts for multiple records combined?*

We certainly acknowledge the limitations of the statistical regime shift analyses and differences between proxy records. However, we do not feel comfortable combining proxy records (e.g. Arctic front proxies and Polar Front proxies) as it may introduce artificial changes that may only arise from the composites. At this point, we prefer to keep the regime shift analyses as they are and re-emphasize the limitations within the Materials and Methods Section 3.5.

*The authors refer to first-order, orbitally driven decrease in northern hemisphere summer and annual insolation as the dominant forcing of the Holocene migration of the Arctic and Polar Fronts (lines 399-406). This section is rather short and it would be useful to explain in more detail how the decrease in insolation led to shifts in the ocean fronts. Furthermore, millennial-scale changes in NADW formation and AMOC strength are briefly mentioned, especially with respect to the SST variability in core JR51-GC35 during the Middle Holocene, but considered as not important control of the progressive migration of the fronts (lines 402-406). However, the frontal proxies in core MD99-2269, including %T. quinqueloba, HBI III and %N. iridea, do reflect millennial-scale variations with rather rapid changes, as depicted by several regime shifts in those records (Fig. 6). Could this not be seen as indication that the migration of the Arctic*

*and Polar Fronts also responded to millennial-scale changes in oceanic and/or atmospheric circulation, for example related to the subpolar gyre circulation? I feel that the discussion of the forcings in section 5.4 should be expanded a little bit.*

We appreciate this suggestion and have expanded the text slightly to more explicitly acknowledge that although there are certainly other controls on higher frequency changes (e.g. NADW, AMOC, NAO), the overarching goal of this paper was to explore the long-term changes reflected in our LOESS-smoothed records. Future work is needed to understand the higher frequency changes in these records and their relation to other atmospheric and oceanic circulation patterns, which we hope to pursue at a later stage.

*Specific comments:*
*Line 18: Perhaps replace "stabilized" by "was located".*
Following Reviewer 1's suggestion, we have edited this to "migrated to".

*Line 20: Perhaps specify that the Arctic and Polar Fronts have moved back "northward" to their current positions.*
Good clarification, which has now been edited.

*Line 37: Revise "advect" ("advects").*
Edited, thank you.

*Lines 95-96: The sedimentation rates and resulting temporal resolution of the proxy records could be mentioned here.*
We agree this is relevant information to include and have edited Section 3.1 accordingly.

*Lines 140-144: The link between the TR25 index and spring phytoplankton blooms as well as the equation for calculating the T25 were established based on samples from the Barents Sea only. A word of caution and a reference to Kolling et al. (2020) (Biomarker distributions in (sub)-Arctic surface sediments and their potential for sea ice reconstructions, G-cubed 21, https://doi.org/10.1029/2019GC008629) may be appropriate regarding the application of this proxy in other regions.*
We discovered this publication after initial submission and agree that it is important to mention the potential complexity of the T25 proxy. However, samples were not included from Iceland, which certainly warrants further investigation with modern samples.

*Line 178: "between select"; This sentence may be revised.*
'Select' has been removed.

*Line 278 and elsewhere: N. iridea should be in italics throughout the manuscript.*
*N. iridea* has now been italicized where mentioned, thank you.

*Line 352: Specify if "radiocarbon reservoir ages" were inferred for surface and/or deep water?*
These radiocarbon ages are based on benthic foraminifera, meaning that they reflect deep water reservoir ages. This has now been clarified in the text, thank you.

*Line 377: "and from toward"; This sentence needs to be revised.*
'From' has now been deleted.

*Fig. 1: Add a reference (Schlitzer, 2020) to the caption for the use of Ocean Data View.*
Reference added, thank you.

*In summary, the manuscript by Harning et al. is well written and contains high quality data that are potentially important new contributions to understanding natural ocean climate variability. I would be happy to recommend publication of this manuscript in Climate of the Past, if the authors address the comments and questions.*
Thank you again for your positive and encouraging review of our manuscript!

---

## Author Response (AR1)

David J. Harning, PhD
Institute of Arctic and Alpine Research
University of Colorado Boulder

3 December 2020

Dear Editor,

Thank you for the invitation to submit our revised manuscript entitled *Response of biological productivity to North Atlantic marine front migration during the Holocene*. In response to your suggestion, our final section of the paper (5.4 Past and future controls on the Holocene migration of the Arctic and Polar Fronts) has been expanded following comments from the peer reviewers. This has now been modified to briefly address the impact of other oceanographic and atmospheric variables, for example NADW, AMOC, and NAO, as well as the movement of marine fronts in other regions of the North Atlantic. While more details of these comparisons are certainly warranted, our main aim in this paper was to compare long-term trends observable in our LOESS smoothing and statistical analyses. Once we finish analyzing the full Deglacial and Holocene portions of these marine sediment records, one major goal in this work will be to explore the shorter term variability of changes in all proxies and how this relates to major climatic and oceanographic events both locally and elsewhere in the North Atlantic. Hence, we prefer to retain these details for a subsequent manuscript.

Please find our revised manuscript with tracked changes attached and thank you again for handling our submission through the review process.

Sincerely,

David J. Harning

---

## Author Response (AR2)

David J. Harning, PhD
Institute of Arctic and Alpine Research
University of Colorado Boulder

22 December 2020

Dear Editor,

Following your suggestions, we are submitting a revised manuscript of *Response of biological productivity to North Atlantic marine front migration during the Holocene*. Below we list each of the specific comments followed by our response. Please refer to the attached revised manuscript for tracked changes.

L41: Consider updating the reference use and clarify. As you say later, the Polar front separate the Polar and Arctic water, while the Arctic front separate the Arctic and Atlantic water; the Arctic water is the domain in between the Atlantic and Polar water - resulting from intermixing between the two end domains. Now it reads like both fronts are between Atlantic water and either Arctic or Polar water.

After re-reading, we agree that this appears unclear and perhaps misleading. We have reordered the sentences so that we immediately define which fronts separate which respective currents before leading into the introduction of the North Iceland Shelf and its proximity to both.

L80: These might be relevant; new detailed knowledge on the physical oceanography of the North Iceland area:
The emergence of the North Icelandic Jet and its evolution from northeast Iceland to Denmark Strait. S Semper, K Våge, RS Pickart, H Valdimarsson, DJ Torres, S Jónsson. Journal of Physical Oceanography 49 (10), 2499-2521
The Iceland-Faroe Slope Jet: a conduit for dense water toward the Faroe Bank Channel overflow S Semper, RS Pickart, K Våge, KMH Larsen, H Hátún, B Hansen. Nature Communications 11 (1), 1-10
Several other papers related to the area, involving Våge/Pickard and others, have been published over the last years.

Thank you for these suggestions! We have added both of the Semper papers to the references and also included the IFSJ in Figure 1.

L187: After Belt or Cabedo-Sanz? Clarify the use of references here.

We have removed the Belt et al. (2019) reference as it the same methods are described in Cabedo-Sanz et al. (2016). Thank you for pointing out this redundancy.

L219: How similar/dissimilar are the new and old counts of T.q. and NPS? You provide number of samples counted, however, how densely are the cores sampled - for the different proxies? Make sure is clear if the analysis presented for different are from the exact same samples and of the same resolution, or if there are differences.

The old TQ and NPS records (Cabedo-Sanz et al., 2016) had high-amplitude variability which was subsequently discovered to have resulted from counting errors. These have all been

recounted, which now removes this variability, resulting in considerably more realistic paleoceanographic interpretations. As for the sample resolution, both planktic and benthic assemblages are multi-decadal. Although the benthic record has more samples overall, all planktics are paired with benthics so that they are directly comparable. We have edited the corresponding text to help clarify these points.

L244: What is the difference in resolution between the two cores? What about differing regime shifts between different proxies from the same crore - or do you have exactly the same samples analyzed for all proxies for each core? Is the resolution the same for each proxy; new and previously published? Both reviewers raised concerns with respect to the results of the statistical analysis. I would like to see a bit more information specifying the limitations more clearly, and argumentation for why you still consider the analysis worthwhile/for what purpose they are useful (and why) and what they don't help you to solve (and why).

Thank you for bringing up this point again. You are right that the varying resolution between proxies within a single core will limit comparison between regime shift analyses, as well as the different cores, as we have pointed out. We hope that our recent additions and edits to this section help clarify any lingering uncertainties for the reader, and make the interpretations of our regime shift analyses more transparent.

L498: What other records/proxies?

We have added MD99-2266 (Moossen et al., 2015) as another $U^{K'}_{37}$ record that shows an increase in SSTs over the last millennium.

L506: From where should this freshwater lid arise? What is the source for the fresh water?

We realize that perhaps freshwater is not the appropriate term as we are really referring to low-salinity Polar Water, which is sourced from East Greenland and well documented in our proxy records. This has now been edited accordingly.

L595: Please add the link to the data.

Unfortunately, we cannot add a direct link at this time. PANGAEA data submissions are currently back-logged. Our dataset is currently in queue and is estimated to be publicly available in ~2 months.

Thank you again for handling our submission through the review process.

Sincerely,

David J. Harning

---

## Author Response (AR3)

David J. Harning, PhD
Institute of Arctic and Alpine Research
University of Colorado Boulder

4 January 2021

Dear Editor,

Following your suggestions, we are submitting a revised manuscript of *Response of biological productivity to North Atlantic marine front migration during the Holocene*. In the introduction we have now added a new reference (Våge et al., 2013) to help reflect the continued and more recent research on modern oceanography north of Iceland.

Sincerely,

David J. Harning